# Non-equispaced Fourier Neural Solvers for PDEs

## Abstract

Solving partial differential equations is difficult. Recently proposed neural resolution-invariant models, despite their effectiveness and efficiency, usually require equispaced spatial points of data. However, sampling in spatial domain is sometimes inevitably non-equispaced in real-world systems, limiting their applicability. In this paper, we propose a Non-equispaced Fourier PDE Solver (NFS) with adaptive interpolation on resampled equispaced points and a variant of Fourier Neural Operators as its components. Experimental results on complex PDEs demonstrate its advantages in accuracy and efficiency. Compared with the spatially-equispaced benchmark methods, it achieves superior performance with $42.85\%$ improvements on MAE, and is able to handle non-equispaced data with a tiny loss of accuracy. Besides, to our best knowledge, NFS is the first ML-based method with mesh invariant inference ability to successfully model turbulent flows in non-equispaced scenarios, with a minor deviation of the error on unseen spatial points.

## 1 Introduction

Solving the partial differential equations (PDEs) holds the key to revealing the underlying mechanisms and forecasting the future evolution of the systems. However, classical numerical PDE solvers require fine discretization in spatial domain to capture the patterns and assure convergence. Besides, they also suffer from computational inefficiency. Recently, data-driven neural PDE solvers revolutionize this field by providing fast and accurate solutions for PDEs. Unlike approaches designed to model one specific instance of PDE (E & Yu, 2017; Bar & Sochen, 2019; Smith et al., 2020; Pan & Duraisamy, 2020; Raissi et al., 2020), neural operators (Guo et al., 2016; Sirignano & Spiliopoulos, 2018; Bhatnagar et al., 2019; KHOO et al., 2020; Li et al., 2020b;c; Bhattacharya et al., 2021; Brandstetter et al., 2022; Lin et al., 2022) directly learn the mapping between infinite-dimensional spaces of functions. They remedy the mesh-dependent nature of the finite-dimensional operators by producing a single set of network parameters that may be used with different discretizations.

However, two problems still exist – discretization-invariant modeling for *non-equispaced data* and *computational inefficiency* compared with convolutional neural networks in the finite-dimensional setting. To alleviate the first problem, MPPDE (Brandstetter et al., 2022) lends basic modules in MPNN (Gilmer et al., 2017) to model the dynamics for spatially non-equispaced data, but even intensifies the time complexity due to the pushforward trick and suffers from unsatisfactory accuracy in complex systems (See Fig. 2(a)). FNO (Li et al., 2020c) has achieved success in tackling the second problem of inefficiency and inaccuracy, while the spatial points must be equispaced due to its harnessing the fast Fourier transform (FFT).

To sum up, two properties should be available in neural PDE solvers: *(1)* discretization-invariance and *(2)* equispace-unnecessity. Property *(1)* is shared by infinite-dimensional neural operators, in which the learned pattern can be generalized to unseen meshes. By contrast, classical vision models and graph spatio-temporal models are not discretization-invariant. Property *(2)* means that the model can handle irregularly-sampled spatial points. For example, graph spatio-temporal models do not require the data to be equispaced, but vision models are equispace-necessary, and limited to handling images as 2-d regular grids. And recently proposed methods can be classified into four types according to the two properties, as shown in Fig. 1. As discussed, although the equispace-necessary methods enjoy fast parallel computation and low prediction error, they lack the ability to handle the spatially non-equispaced data. For these reasons, this paper aims to design a mesh-invariant model (defined in Fig. 1) called Non-equispaced Fourier neural Solver (NFS) with comparably low cost of computation

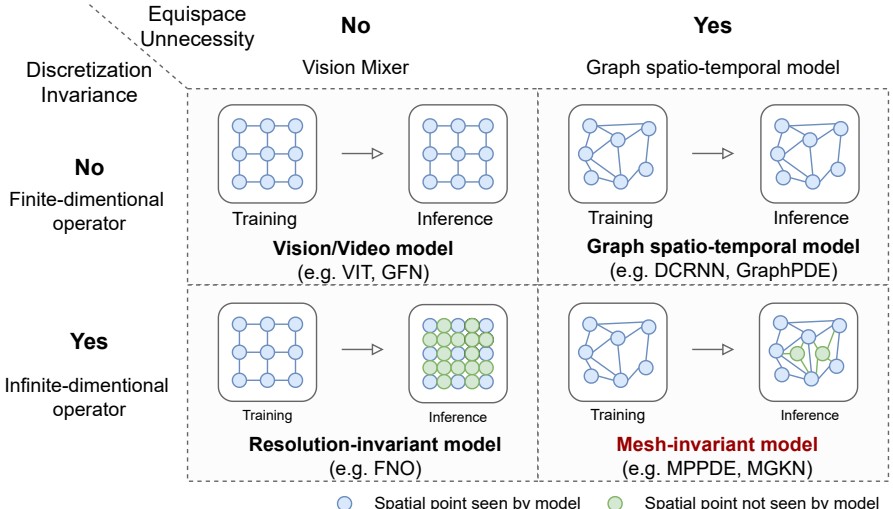

Figure 1: Four types of methods with or without the two concluded limitations.

and high accuracy, by lending the powerful expressivity of FNO and vision models to efficiently solve the complex PDE systems. Our paper including leading contributions is organized as follows:

- In Sec. 2, we first give some preliminaries on neural operators as related work, with a brief introduction to Vision Mixers, to build a bridge between Fourier Neural Operator and Vision Mixers. Thus, we illustrate our motivation for the work: To establish a mesh-invariant neural operator, by harnessing the network structure of Vision Mixers.

- In Sec. 3, we proposed a Non-equispaced Fourier Solver (NFS), with adaptive interpolation operators and a variant of Fourier Neural Operators as the components. Approximation theorems that guarantee the expressiveness of the proposed interpolation operators are developed. Further discussion gives insights into the relation between NFS, patchwise embedding and multipole graph models.

- In Sec. 4, extensive experiments on different types of PDEs are conducted to demonstrate the superiority of our methods. Detailed ablation studies show that both the proposed interpolation kernel and the architecture of Vision Mixers contribute to the improvements in performance.

## 2 BACKGROUND AND RELATED WORK

### 2.1 PROBLEM STATEMENT

Let $D \in \mathbb{R}^d$ be the bounded and open spatial domain where $n_s$-point discretization of the domain $D$ written as $\boldsymbol{X} = \{\boldsymbol{x}_i = (x_i^{(1)}, \ldots, x_i^{(d)}) : 1 \leq i \leq n_s\}$ are sampled. The observation of input function $a \in \mathcal{A}(D; \mathbb{R}^{d_a})$ and output $u \in \mathcal{U}(D; \mathbb{R}^{d_u})$ on the $n_s$ points are denoted by $\{a(\boldsymbol{x}_i), u(\boldsymbol{x}_i)\}_{i=1}^{n_s}$, where $\mathcal{A}(D; \mathbb{R}^{d_a})$ and $\mathcal{U}(D; \mathbb{R}^{d_u})$ are separable Banach spaces of function taking values in $\mathbb{R}^{d_a}$ and $\mathbb{R}^{d_u}$ respectively. Suppose $\boldsymbol{x} \sim \mu$ is i.i.d. sampled from the probability measure $\mu$ supported on $D$. An infinite-dimensional neural operator $\mathcal{G}_\theta : \mathcal{A}(D; \mathbb{R}^{d_a}) \to \mathcal{U}(D; \mathbb{R}^{d_u})$ parameterized by $\theta \in \Theta$, aims to build an approximation so that $\mathcal{G}_\theta(a) \approx u$. A cost functional $\mathcal{C} : \mathcal{U}(D; \mathbb{R}^{d_u}) \times \mathcal{U}(D; \mathbb{R}^{d_u}) \to \mathbb{R}$ is defined to optimize the parameter $\theta$ of the operator by the objective

$$\min_{\theta \in \Theta} \mathbb{E}_{\boldsymbol{x} \sim \mu}[\mathcal{C}(\mathcal{G}_\theta(a), u)(\boldsymbol{x})] \approx \frac{1}{n_s} \sum_{i=1}^{n_s} \mathcal{C}(\mathcal{G}_\theta(a), u)(\boldsymbol{x}) \tag{1}$$

To establish a mesh-invariant operator, $\boldsymbol{X}$ can be non-equispaced, and the learned $\mathcal{G}_\theta$ should be transferred to an arbitrary discretization $\boldsymbol{X}' \in D$, where $\boldsymbol{x} \in \boldsymbol{X}'$ can be not necessarily contained in $\boldsymbol{X}$. Because we focus on spatially non-equispaced points, when the PDE system is time-dependent, we assume that timestamps $\{t_j\}$ are uniformly sampled, which means we do not focus on temporally irregular sampling or continuous time problem (Rubanova et al., 2019; Chen et al., 2019; Çağatay Yıldız et al., 2019; Iakovlev et al., 2020).

## 2.2 DISCRETE FOURIER TRANSFORM

Let $\boldsymbol{k}_l = (k_l^{(1)}, \ldots, k_l^{(d)})$ the $l$-th frequency corresponding to $\boldsymbol{X}$, with $\boldsymbol{k}_l \in \mathbb{Z}^d$. The discrete Fourier transform of $f : D \to \mathbb{R}^{d_f}$ is denoted by $\mathcal{F}(f)(\boldsymbol{k}) \in \mathbb{C}^{d_f}$, with $\mathcal{F}^{-1}$ as its inverse, then

$$\mathcal{F}(f)^{(j)}(\boldsymbol{k}_l) = \sum_{i=1}^{n_s} f^{(j)}(\boldsymbol{x}_i) e^{-2i\pi <\boldsymbol{x}_i, \boldsymbol{k}_l>}, \qquad \mathcal{F}^{-1}(f)^{(j)}(\boldsymbol{x}_i) = \sum_{l=1}^{n_s} f^{(j)}(\boldsymbol{k}_l) e^{2i\pi <\boldsymbol{x}_i, \boldsymbol{k}_l>}, \quad (2)$$

where $j$ means the $j$-th dimension of $f$. General Fourier transforms have complexity $O(n_s^2)$. When the spatial points are distributed uniformly on equispaced grids, fast Fourier transform (FFT) and its inverse (IFFT) (Rader & Brenner, 1976) can be implemented to reduce the complexity to $O(n_s \log n_s)$.

## 2.3 FOURIER NEURAL OPERATOR

**Neural Operators.** To model one specific instance of PDEs, a line of neural solvers have been designed, with prior physical knowledge as constraints. Different from these methods, neural operators (Lu et al., 2021; Nelsen & Stuart, 2021) require no knowledge of underlying PDEs, and only data. Finite-dimensional operator methods (Guo et al., 2016; Sirignano & Spiliopoulos, 2018; Bhatnagar et al., 2019; KHOO et al., 2020) are discretization-variant, meaning that the model can only learn the patterns of the spatial points which have been fed to the model in the training process. By contrast, infinite-dimensional operator methods (Li et al., 2020b;c; Bhattacharya et al., 2021; Brandstetter et al., 2022) are proposed to be discretization-invariant, enabling the learned models to generalize well to unseen meshes with zero-shot.

**Kernel integral operator method** (Li et al., 2020a) is a family of infinite-dimensional operators, in which $(\mathcal{G}_\theta(a))(\boldsymbol{x}) = Q \circ v^{\mathrm{T}} \circ \cdots \circ v^1 \circ P(a)(\boldsymbol{x})$ is formulated as an iterative architecture. A higher-dimensional representation function is first obtained by $v^0 = P(a) \in \mathcal{U}(D; \mathbb{R}^{d_v})$, where $P$ is a shallow fully-connected network. It is updated by

$$v^{\mathrm{t}+1}(\boldsymbol{x}) := \sigma(Wv^{\mathrm{t}}(\boldsymbol{x}) + \mathcal{K}_\phi(a)v^{\mathrm{t}}(\boldsymbol{x})), \qquad \forall \boldsymbol{x} \in D \qquad (3)$$

where $\mathcal{K}_\phi : \mathcal{A} \to \mathcal{L}(\mathcal{U})$ is a kernel integral operator mapping, mapping $a$ to bounded linear operators, with parameters $\phi$. $W$ is a linear transform and $\sigma$ is a non-linear activation function. After the final iteration, $Q$ projects $v^{\mathrm{T}}(\boldsymbol{x})$ back to $\mathcal{U}(D; \mathbb{R}^{d_u})$.

**Fourier Neural Operator** (FNO) (Li et al., 2020c) as a member in kernel integral operator methods, updates the representation by applying the convolution theorem as:

$$\mathcal{K}_\phi(a)v(\boldsymbol{x}) = \mathcal{F}^{-1}(\mathcal{F}(\kappa_\phi) \cdot \mathcal{F}(v))(\boldsymbol{x}) = \mathcal{F}^{-1}(R_\phi \cdot \mathcal{F}(v))(\boldsymbol{x}), \qquad (4)$$

where $R_\phi$ as the Fourier transform of a periodic kernel function $\kappa_\phi$, is directly learned as the parameters in the updating process. To be resolution-invariant, FNO picks a finite-dimensional parameterization by truncating the Fourier series of both $\mathcal{F}(v)$ and $R_\phi$ as a maximal number of modes $k_{\max}^{(l)}$ for $1 \le l \le d$. Because the sampled spatial points are equispaced in FNO, it can conduct FFT and IFFT to get the Fourier series, which can be very efficient.

## 2.4 VISION MIXER AND GRAPH SPATIO-TEMPORAL MODEL

**Vision Mixers** (Tolstikhin et al., 2021; Rao et al., 2021; Guibas et al., 2021) are a line of models with a stack of (token mixing) - (channel mixing) - (token mixing) as their network structure for vision tasks. They are based on the assumption that the key component for the effectiveness of Vision Transformers (VIT) (Dosovitskiy et al., 2020) is attributed to the proper mixing of tokens. The defined tokens are equivalent to equispaced spatial points in the former definition, and the research on the mixing of them can be an analogy to modeling the proper interaction or message-passing patterns among spatial points. In specific, VIT uses a non-Mercer kernel function (Wright & Gonzalez, 2021) $\kappa_\phi$ to adaptively learn the pattern of message-passing through the iterative updating process

$$v^{\mathrm{t}+1}(\boldsymbol{x}) = \sigma(\mathrm{ChannelMix} \circ \mathrm{TokenMix}(v^{\mathrm{t}}(\boldsymbol{x})));$$

$$\mathrm{TokenMix}(v(\boldsymbol{x})) = \sum_i \kappa_\phi(\boldsymbol{x}, \boldsymbol{x}_i, v(\boldsymbol{x}), v(\boldsymbol{x}_i)) \cdot v(\boldsymbol{x}_i); \quad \mathrm{ChannelMix}(v(\boldsymbol{x})) = Wv(\boldsymbol{x}), \qquad (5)$$

where $W$ is a linear transform called channel mixing layer because it transforms the input on the channel of an image whose dimension is equivalent to function dimension $d_f$. Note that we omit the residual connection in Eq. (3) for simplicity.

**Remark.** The FNO can be regarded as a member of the family of Vision Mixers. The reason is that a component in an iteration in Eq. (4) can be written as $(R_\phi \cdot \mathcal{F}(v))(\boldsymbol{x}) = R_\phi \cdot \sum_i e^{-2\pi i < \boldsymbol{x}, \boldsymbol{x}_i >} v(\boldsymbol{x}_i)$, because in the equispaced scenarios, $\boldsymbol{x}_i$ can be regarded as lying on the same grids as $\boldsymbol{k}$ after scaling. The kernel $\kappa_\phi$ is parameterized by $\kappa_\phi(\boldsymbol{x}, \boldsymbol{x}_i, v(\boldsymbol{x}), v(\boldsymbol{x}_i)) = e^{-2\pi i < \boldsymbol{x}, \boldsymbol{x}_i >}$, and the matrix multiplication of $R_\phi$ also performs mixing on channels. Besides, the inverse Fourier transform can also be regarded as token mixing layers, or so-called token demixing layers (Guibas et al., 2021).

However, the powerful fitting ability and efficiency of Vision Mixers are limited to being applied to non-equispaced spatial points. Another option for non-equispaced data is graph spatio-temporal models, in which interaction patterns among spatial points are modeled in a graph message-passing way (Gilmer et al., 2017; Atwood & Towsley, 2016; Defferrard et al., 2017). The mechanism is similar to the token mixing in Vision Mixers by means of the summation in Eq. (5) conducted in the predefined neighborhood of each point. Unfortunately, the graph spatio-temporal models (Seo et al., 2016; Li et al., 2018; Bai et al., 2020; Lin et al., 2021) suffer from high computational complexity and unsatisfactory accuracy in solving complex dynamical systems (such as turbulent flows).

## 2.5 MOTIVATION

Since FNO belongs to Vision Mixers, this firstly raised a question to us: *Do models employing Vision Mixers's architecture have the potential to model complex PDE systems?* Thus, experiments are conducted to give an intuitive explanation of our motivation as shown in Fig. 2. The data are generated by Navier-Stokes equations. It is noted that graph spatio-temporal methods can also handle the equispaced data. Detailed setup is given in Sec. 4.

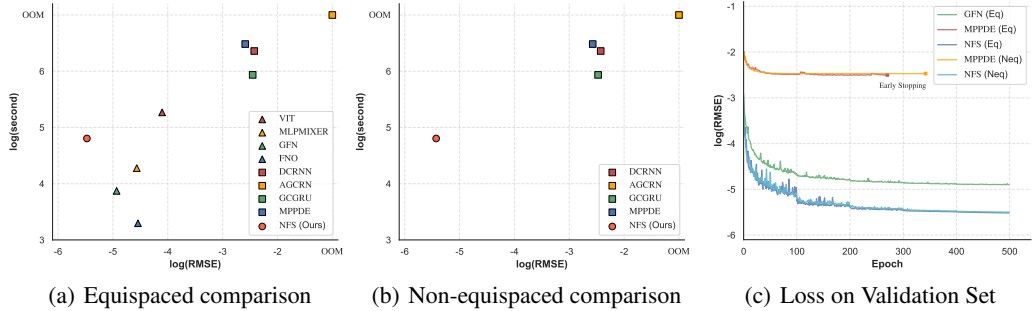

|  |  |  |
|:-:|:-:|:-:|
| (a) Equispaced comparison | (b) Non-equispaced comparison | (c) Loss on Validation Set |

Figure 2: Intuitive explanation of our motivation: In (a) and (b), '△' represents Vision Mixers, '□' represents graph spatio-temporal models and '○' is the proposed NFS. In (c), 'Eq' and 'Neq' mean the methods are trained in equispaced and non-equispaced scenarios respectively.

We find that *(1)* All of the evaluated Vision Mixers are able to model the dynamical systems effectively, in spite of FNO as the only discretization-invariant model; *(2)* The complex dynamics of the systems are hardly captured by graph spatio-temporal models, whose performance on both accuracy and computational efficiency is very unsatisfactory in either equispaced or non-equispaced scenarios. Fig. 2(c) shows the problem of infeasibility of graph spatio-temporal models through the loss curves on the validation set, compared with Vision Mixers. However, Vision Mixers fail to handle the non-equispaced data. Therefore, we aim to *(1)* establish a **mesh-invariant** model, by harnessing the network structure of Vision Mixers to achieve competitive **efficiency** and **effectiveness** in equispaced scenarios, as shown in Fig. 2(a); *(2)* Besides, it should allow applicability and comparable accuracy in **non-equispaced** scenarios for solving PDEs, as shown in Fig. 2(b).

## 3 PROPOSED METHOD

### 3.1 NON-EQUISPACED FOURIER TRANSFORM

Nonuniformly signals are unavoidable in certain real-world physics scenarios, such as signals obtained by meteorological stations on the earth surface, which urge the fast Fourier transform (FFT) to be extended to non-equispaced data with efficient implementation of FFT. Non-equispaced FFTs usually rely on a mixture of interpolation and the judicious use of FFT, where the calculations of interpolation are no more than $O(n_s \log n_s)$ operations (Kalamkar et al., 2012; Cheema et al., 2017). For example, Lagrange interpolation is used to approximate the signal values on $m_s$ resampled equispaced points

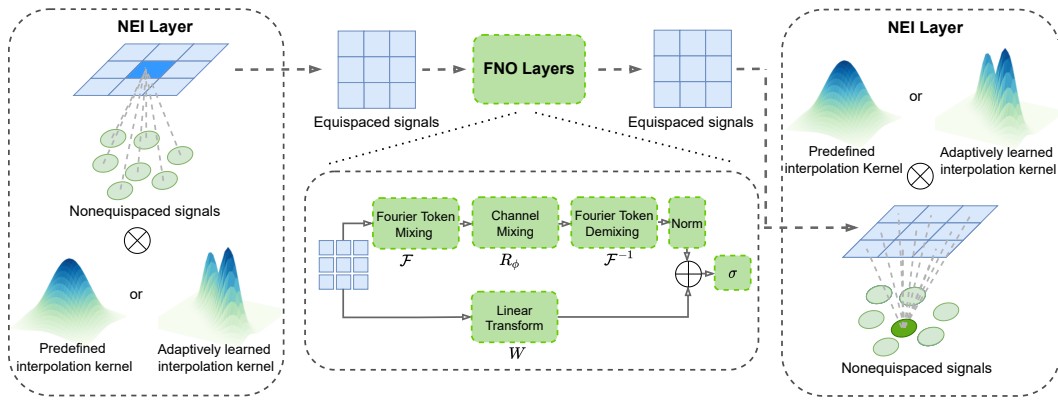

Figure 3: The architecture of NFS: In non-equispaced inperpolation (NEI) layers, the interpolation kernels are adaptively learned rather than predefined, and the interpolated equispaced signals are processed through a stack of FNO layers with the same structure of Vision Mixers.

$\{\boldsymbol{x}_j\}_{1 \leq j \leq m_s}$, and then implement FFT on the interpolated points. A low rank approximation with complexity of $O(n_s \log(1/\epsilon))$ is used to replace the interpolation with complexity of $O(n_s^2)$ with $\epsilon$ as the precision of computations (Dutt & Rokhlin, 1995). Another example is commonly used Gaussian-based interpolation (Kestur et al., 2010). Denote $\mathcal{F}$ as equispaced FFT in particular, and $\mathcal{H}$ as the interpolation operator, and the proposed non-equispaced FFT can be written as

$$(\mathcal{F} \circ \mathcal{H}(f))(\boldsymbol{k}) \approx \sqrt{\frac{\pi}{\tau}} e^{\tau < \boldsymbol{k}, \boldsymbol{k} >} \sum_{j=1}^{m_s} e^{-2i\pi < \boldsymbol{k}, \boldsymbol{x}_j >} \sum_{i=1}^{n_s} f(\boldsymbol{x}_i) h_\tau(\boldsymbol{x}_i - \boldsymbol{x}_j). \tag{6}$$

$\mathcal{H}(f)(\boldsymbol{x}_j) = \sum_{i=1}^{n_s} f(\boldsymbol{x}_i) h_\tau(\boldsymbol{x}_i - \boldsymbol{x}_j)$ interpolates values on resampled points via convolution with the periodic heat kernel $h_\tau(\boldsymbol{x} - \boldsymbol{y}) = \sum_{\boldsymbol{l} \in \mathbb{Z}^d} e^{-(\boldsymbol{x}-\boldsymbol{l})^2/4\tau}$, with $\tau$ as a constant. Multiplication of the inperploation matrix $(H_\tau)_{i,j} = h_\tau(\boldsymbol{x}_i - \boldsymbol{x}_j)$ and the signal vector $(\boldsymbol{f})_i = f(\boldsymbol{x}_i)$ includes $O(n_s m_s) \approx O(n_s^2)$ operations. For the kernel $h_\tau$, it is a summation of Gaussian kernel, and convolution with a single Gaussian in each points $\boldsymbol{x}_i$'s neighborhood $\mathcal{N}(\boldsymbol{x}_i)$ can yeid a tiny error depending on $\tau$, so the interpolation operator can be approximate via $\mathcal{H}(f)(\boldsymbol{x}_j) = \sum_{\boldsymbol{x}_j \in \mathcal{N}(\boldsymbol{x}_i)} f(\boldsymbol{x}_i) h_\tau(\boldsymbol{x}_i - \boldsymbol{x}_j)$. Restricting the neighbor number to $|\mathcal{N}(\boldsymbol{x}_i)| \leq \log n_s$ leads the complexity to reduce to $O(n_s \log n_s)$.

### 3.2 NON-EQUISPACED FOURIER NEURAL PDE SOLVER

**Non-equispaced interpolation.** To harness the effectiveness of FNO, we use non-equispaced Fourier token mixing instead of the equispaced one. It generalizes the equispaced FFT in Eq. (4) as

$$\tilde{\mathcal{F}}(v) = (\mathcal{F} \circ \mathcal{H}_\eta(a))(v). \tag{7}$$

We denote $\mathcal{H}_\eta : \mathcal{A} \to \mathcal{L}(\mathcal{U})$ as the interpolation operator mapping, which maps parametric function to a bounded interpolation operator. $\mathcal{H}_\eta(a)$ gets the inerploated values on $m_s$ resampled equispaced points via the convolution with kernel $h_\eta$ as

$$(\mathcal{H}_\eta(a)v)(\boldsymbol{x}_j) = \frac{1}{n_s} \sum_{i=1}^{n_s} v(\boldsymbol{x}_i) h_\eta(\boldsymbol{x}_j - \boldsymbol{x}_i, \boldsymbol{x}_i, a(\boldsymbol{x}_i)), \tag{8}$$

where $\boldsymbol{x}_j$ lies on resampled equispaced grids. Another $\mathcal{H}'_\zeta$ interpolates back on the $n_s$ non-equispaced ones in the same way via the convolution with kernel $h_\zeta$. To reduce the operations to no more than $O(n_s \log n_s)$, the summation is restricted in the neighborhood of $\boldsymbol{x}_i$ and $\boldsymbol{x}_j$, such that $|\mathcal{N}(\boldsymbol{x}_i)| \approx |\mathcal{N}(\boldsymbol{x}_j)| \leq c \log n_s$ with $c$ as a predefined constant determining the neighborhood size of spatial points. We formulate the kernel with a shallow feed-forward neural networks. Thanks to the universal approximation of neural networks, the following theorem assures that the interpolation operator can approximate the representation function $v$ arbitrarily well. (For detailed proof, see Appendix. A.3.) Empirical observations on the convergence of interpolation operators are given in Appendix C.

**Theorem 3.1** (Approximation Theorem of the Adaptive Interpolation). *Assume the setting of **Theorem A2** in Appendix. A.3 is satisfied. $\mu$ is the probability measure supported on $D$. For $v \in \mathcal{U}$, suppose $\mathcal{U} = L^p(D; \mathbb{R}^{d_v})$, for any $1 < p < \infty$. Then, given $\epsilon > 0$, there exist a neural network $h_\eta : \mathbb{R}^d \times \mathbb{R}^d \times \mathbb{R}^{d_a} \to \mathbb{R}^{d_v}$, such that $||\hat{v} - v||_{\mathcal{U}} \le \epsilon$, where $\hat{v}(\boldsymbol{x}) = \int_D h_\eta(\boldsymbol{x} - \boldsymbol{y}, \boldsymbol{x}, a(\boldsymbol{y})) v(\boldsymbol{y}) d\mu(\boldsymbol{y})$.*

**Applicability of Layer-Norm.** As shown in Fig. 3, besides the comparison of the proposed interpolation operator with the traditional ones, a notable difference between the original FNO and FNO layers in our Vision Mixer architecture is the applicability of normalization layers (Layer-Norm) which is usually used in Vision Mixers' architecture. FNO cannot adapt Layer-Norm layers, because the change of resolution will make the trained normalization parameters and spatial points disagree with each other. In comparison, the resampled equispaced points are fixed in our architecture, no matter how the discretization of the input changes. Therefore, the normalization layers can be added, in a similar way to Vision Mixers, bringing considerable improvements (See Sec. 4.3).

**Mesh invariance.** In the intermediate layers, which adopt equispaced FNO, the resampled points are fixed in both training and inference process, invariant to the input meshes. In the interpolation layers, the operator $\mathcal{H}_\eta(a)$ is discretization-invariant because the kernel can be inductively obtained by the newly observed signals $a(\boldsymbol{x})$, its coordinate $\boldsymbol{x}$ and resampled spatial points' relative coordinates $\boldsymbol{x}_j - \boldsymbol{x}$. In the same way, $\mathcal{H}'_\zeta(a)$ is also mesh-invariant. This allows the NFS to achieve zero-shot mesh-invariant inference ability, which is demonstrated in Sec. 4.2.

**Complexity analysis.** The complexity of FNO is $O(n_s \log n_s + n_s k_{\max})$. In the interpolation layers, because the interpolated values of resampled points are determined by their neighbors, we set the size of each resampled point's neighborhood in $\mathcal{G}$ and observed non-equispaced points's neighborhood in $\mathcal{G}'$ as $|\mathcal{N}(\boldsymbol{x}_i)| \approx |\mathcal{N}(\boldsymbol{x}_j)| \le c \log(n_s)$, for $1 \le i \le n_s, 1 \le j \le m_s$. And in this way, the sparsity of the interpolation matrix reduces the complexity of the two interpolation layers to $O(c \cdot n_s \log n_s + c \cdot m_s \log n_s)$. If we set the resampled points number as $n_s$, the overall complexity is $O(2c \cdot n_s \log n_s + n_s \log n_s + n_s k_{\max}) \sim O(n_s \log n_s + n_s k_{\max})$.

### 3.3 FURTHER DISCUSSION

**Relation to Vision Mixer.** The interpolation can be compared to patchwise embedding in Vision Mixers. For example, MLPMIXER learns the token mixing patterns adaptively with a feed-forward network, but the high resolution of input images does not permit the global mixing of tokens due to the complexity of $O(n_s^2)$. Therefore, the input images are firstly rearranged into patches, with each patch containing $n_p$ pixels. In this way, the complexity is reduced to $O(n_s^2/n_p^2)$, enabling feasible token mixing. The patchwise embedding is very similar to interpolating the values on resampled points, as the former one first chooses patches' centers as $n_s^2/n_p^2$ resampled points, and 'interpolates' the resampled points by lifting the embedding dimension and the rearranging of their neighbors' values as the interpolated values, rather than using a kernel.

**Relation to multipole graph model.** The adaptively learned interpolation layer in NFS has a similar formulation of multipole graph models (Li et al., 2020b). In multipole graph models, the high-level nodes aggregate messages from their low-level neighbors as $v^{\mathrm{High}}(\boldsymbol{x}_j) = \frac{1}{|\mathcal{N}(\boldsymbol{x}_j)|} \sum_{\boldsymbol{x}_i \in \mathcal{N}(\boldsymbol{x}_j)} v^{\mathrm{Low}}(\boldsymbol{x}_i) h_\eta(\boldsymbol{x}_j, \boldsymbol{x}_i, a(\boldsymbol{x}_j), a(\boldsymbol{x}_i))$. Compared to multipole graph models, the values of high-level resampled equispaced nodes are approximated with low-level non-equispaced nodes' values in NFS, but nodes' values of low levels are given in multipole graphs. This causes differences in multipole graph models' message-passing and NFS's interpolation: In the former one, messages flow circularly among different levels of nodes, while in NFS, messages only exchange twice between the nodes of two levels – one is from low-level non-equispaced nodes to high-level resampled equispaced nodes, and the other is the opposite.

## 4 EXPERIMENTS

### 4.1 EXPERIEMNTAL SETUP

**Benchmarks for comparison.** For finite-dimensional operators, we choose Vision Mixers including VIT (Dosovitskiy et al., 2020), GFN (Rao et al., 2021) and MLPMIXER (Tolstikhin et al., 2021) as equispaced problem solvers, with DEEPONET-V and DEEPONET-U as two variants for DeepONet(Lu et al., 2021) and graph spatio-temporal models including DCRNN (Li et al., 2018),

AGCRN (Bai et al., 2020) and GCGRU (Seo et al., 2016) as non-equispaced problem solvers. For infinite-dimensional operators, the state-of-the-art FNO (Li et al., 2020c) for equispaced problems and MPPDE (Brandstetter et al., 2022) for non-equispaced problems are chosen. A brief introduction to these models is shown in Appendix. B.1. In Vision Mixers, the different timestamps in temporal axis are also regarded as 'tokens' in that timestamps are uniformly sampled.

**Protocol.** The widely-used metrics - Mean Absolute Error (MAE) and Root Mean Square Error (RMSE) are deployed to measure the performance. The reported mean and standard deviation of metrics are obtained through 5 independent experiments. All the models for comparison are trained with target function of MSE, i.e. $\mathcal{C}(u,v)(\boldsymbol{x}) = ||u(\boldsymbol{x}) - v(\boldsymbol{x})||^2$ corresponding to Eq. (1), and optimized by Adam optimizer in 500 epochs. The hyper-parameters are chosen through a carefully tuning on the validation set. Every trial is implemented on a single Nvidia-V100 (32510MB).

## 4.2 NUMERICAL EXPERIMENTS

**Data.** We choose four equations for numerical experiments, three of which are time-dependent (KdV, Burgers' and NS), while the other one is not (Darcy Flows). For 1-d problem, we consider Korteweg de Vries (KdV) and Burgers' equation (given in Appendix. B.2.).

For 2-d PDEs, we consider Darcy Flow (given in Appendix. B.2.) and Navier-Stokes (NS) equation for a viscous, incompressible fluid in vorticity form on the unit torus:

$$\partial_t w(\boldsymbol{x}, t) + u(\boldsymbol{x}, t) \cdot \nabla w(\boldsymbol{x}, t) = \nu \Delta w(\boldsymbol{x}, t) + f(\boldsymbol{x}),$$
$$\nabla \cdot u(\boldsymbol{x}, t) = 0, \qquad w(\boldsymbol{x}, 0) = w_0(\boldsymbol{x}),$$
(9)

where $\boldsymbol{x} \in [0,1]^2, t \in [0,1]$. $u$ is the velocity field, $w = \nabla \times u$ is the vorticity, $w_0$ is the initial vorticity, $\nu \in \mathbb{R}^+$ is the viscosity coefficient, and $f$ is the forcing function.

The total number of instances is 1200, with percentages of 0.7, 0.1 and 0.2 for training, validating and testing, respectively. The original simulated resolutions of PDE signals are $128 \times 128$ in NS equation. For others, see Appendix. B.2. When evaluating their performance in equispaced scenarios of different resolutions, we can downsample the resolution for training to low-resolution data, e.g. $64 \times 64$ in NS equation. To evaluate their performance in non-equispaced scenarios of different meshes, we randomly choose $n_s$ spatial points for training.

Table 1: MAE($\times 10^{-3}$) comparison with vision mixer benchmarks.

| | Burgers' ($n_t = 10$) | | | Darcy Flow | | | NS ($n_t = 1$) | | | NS ($n_t = 10$) | | |
|---|---|---|---|---|---|---|---|---|---|---|---|---|
| $r$ | 512 | 512 | 1024 | 64 | 128 | 256 | 64 | 64 | 128 | 64 | 64 | 128 |
| $n_t'$ | 10 | 40 | 20 | 1 | 1 | 1 | 10 | 40 | 20 | 10 | 40 | 20 |
| VIT | 0.5042 | 2.4269 | 1.5327 | 0.5073 | 0.9865 | 1.1078 | 9.3797 | 22.8565 | 15.7398 | 3.9609 | 12.3433 | 9.3010 |
| MLPMIXER | 0.1973 | 0.4210 | 0.3303 | 0.4970 | 0.8909 | 0.9125 | 7.5246 | 15.8632 | 14.9360 | 3.1530 | 7.9291 | 7.7410 |
| GFN | 0.2383 | 0.4187 | 0.3500 | 0.4739 | 0.8659 | 0.9618 | 3.5524 | 10.2250 | 6.3976 | 1.7396 | 5.4464 | 3.1261 |
| FNO | 0.0978 | 0.1815 | **0.1430** | 0.4289 | 0.7086 | 0.9075 | 3.3425 | 8.9857 | 4.4627 | 2.4076 | 7.6979 | 3.7001 |
| DEEPONET-U | 0.4471 | 1.9624 | 0.6541 | 0.3753 | 0.9488 | 0.9692 | 7.4912 | 16.0440 | 14.3476 | 3.4436 | 10.2950 | 7.1394 |
| DEEPONET-V | 0.4782 | 2.1707 | 1.6131 | 0.5119 | 0.9614 | 1.3216 | 8.6986 | 18.5561 | 16.0587 | 3.9745 | 12.3314 | 9.3471 |
| NFS | **0.0958** | **0.1708** | 0.1474 | **0.1497** | **0.2254** | **0.4216** | **1.7425** | **4.7882** | **2.6988** | **0.8636** | **3.1122** | **1.8406** |

Table 2: MAE($\times 10^{-3}$) comparison with graph spatio-temporal benchmarks.

| | Burgers' ($n_t = 10$) | | | Darcy Flow | | | NS ($n_t = 1$) | | | NS ($n_t = 10$) | | |
|---|---|---|---|---|---|---|---|---|---|---|---|---|
| $n_s$ | 512 | 512 | 256 | 4096 | 16384 | 1024 | 4096 | 4096 | 1024 | 4096 | 4096 | 1024 |
| $n_t'$ | 10 | 40 | 20 | 1 | 1 | 1 | 10 | 40 | 20 | 10 | 40 | 20 |
| DCRNN | 2.6122 | 8.5880 | 4.6126 | 1.7629 | OOM | 1.8146 | 30.6756 | 88.3382 | 52.1290 | 8.7025 | 59.6602 | 27.1069 |
| AGCRN | 4.6667 | 15.6143 | 10.4900 | 1.7336 | OOM | 1.6938 | OOM | OOM | 59.9393 | OOM | OOM | 42.4197 |
| GCGRU | 1.6643 | 5.7653 | 3.1400 | 1.7403 | OOM | 1.7633 | 28.8537 | 85.9303 | 49.9352 | 6.3570 | 57.2493 | 21.3537 |
| MPPDE | 1.1271 | 4.1213 | 2.4554 | 0.5608 | 0.6384 | 0.6673 | 8.9810 | 54.2387 | 20.7453 | 5.4353 | 42.3057 | 17.5902 |
| NFS | **0.1085** | **0.1983** | **0.1634** | **0.1430** | **0.2379** | **0.1727** | **2.1992** | **4.7865** | **3.9178** | **0.9335** | **3.2768** | **1.8239** |

**Performance comparison.** In this part, for time-dependent PDEs, our target is to map the observed physical quantities from initial condition $u(\boldsymbol{X}, \boldsymbol{T}) \in \mathbb{R}^{n_s \times n_t}$, where $\boldsymbol{T} = \{t_i : t_i < T\}_{1 \le i \le n_t}$, to quantities at some later time $u(\boldsymbol{X}, \boldsymbol{T'}) \in \mathbb{R}^{n_s \times n_t'}$, where $\boldsymbol{T'} = \{t_i : T < t_i < T'\}_{1 \le i \le n_t'}$. We set the input timestamp number $n_t$ as 1 (initial state to future dynamics) and 10 (sequence to sequence), and prediction horizon $n_t'$ as 10, 20 and 40 as short-, mid- and long-term settings. For Darcy Flows, which are independent of time, we directly build an operator to map $a$ to $u$. In equispaced scenarios, the resolution is denoted by $r^d = n_s$, where $d$ is the spatial dimension. In non-equispaced scenarios,

the spatial points number is denoted by $n_s$. The comparison of benchmarks with or without equispace-unnecessity are shown in Table. 1 and 2 respectively, and detailed results including KdV equations with RMSE and standard deviations are given in Appendix. B.3. It can be concluded that *(1)* In equispaced scenarios, the proposed NFS obtains the lowest error in most 1-d PDE settings, and in solving 2-d PDEs, its superiority over other Vision Mixers are significant, with $42.85\%$ improvements on MAE according to the trials of NS ($r = 64, n_t = 10, n_t' = 40$). *(2)* In non-equispaced scenarios, the evaluated graph spatio-temporal models' performance is unsatisfactory, especially in NS equations. In comparison, NFS achieves comparable high accuracy to the equispaced scenarios, for instance, according to columns of NS ($r = 64, n_t = 10, n_t' = 40$) with ($n_s = 4096, n_t = 10, n_t' = 40$). *(3)* In some trials such as Burgers' ($n_t = 1$) in Table. B4 in Appendix. B.3, Vison Mixers including FNO also suffer from non-convergence of loss; while NFS can still generate accurate predictions. The explanation of the phenomenon will be our future work.

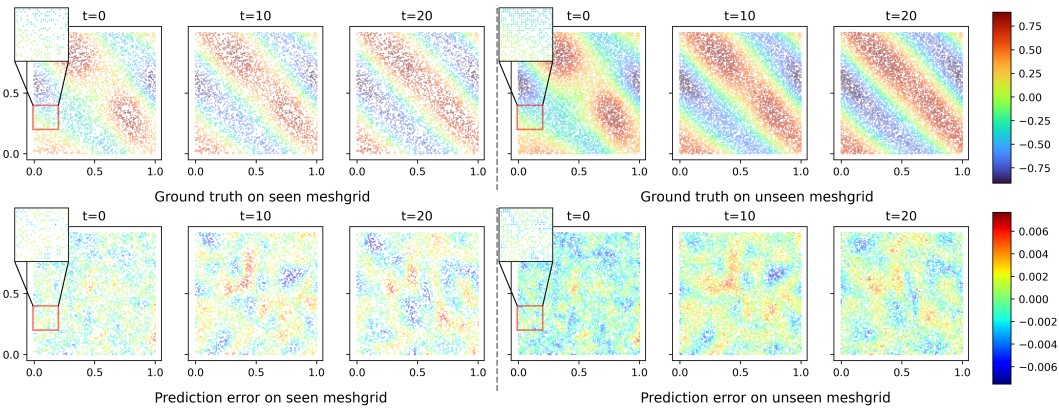

Figure 4: Visualization on non-equispaced NS equation: The training mesh ($n_s = 4096$) differs from the mesh in inference process ($n_s' = 8192$). Appendix. B.4 gives more visualization.

Table 3: Results on NS equation: MAE($\times 10^{-3}$) of NFS and different variants of NFS on seen and unseen meshes. 'Flex + LN' is the proposed NFS. 'Flex' represents the flexible interpolation layer defined in Eq. (8), 'LN' is the Layer-Norm and 'Gaus' is the predefined Gaussian interpolation. Appendix. B.5 gives details and results on other equations.

|  | ($n_s = 4096, n_t = 10, n_t' = 10$) | | | ($n_s = 1024, n_t = 10, n_t' = 20$) | | | ($n_s = 4096, n_t = 10, n_t' = 40$) | | |
|---|---|---|---|---|---|---|---|---|---|
| $n_s'$ | Flex + LN | Gaus + LN | Flex + LN̸ | Flex + LN | Gaus + LN | Flex + LN̸ | Flex + LN | Gaus + LN | Flex + LN̸ |
| $n_s$ | 0.9335 | 1.6341 | 1.2138 | 1.8239 | 2.1976 | 2.5119 | 3.2768 | 3.6422 | 5.6761 |
| $2n_s$ | 0.9731 | 2.8589 | 1.4882 | 2.3530 | 3.7465 | 7.0203 | 3.5439 | 3.9092 | 5.8975 |
| $3n_s$ | 1.1071 | 3.4513 | 1.6384 | 2.5179 | 5.7712 | 7.9177 | 3.6584 | 4.2102 | 6.6622 |
| $4n_s$ | 1.1015 | 3.4357 | 1.6975 | 2.5919 | 5.5990 | 7.1962 | 3.6608 | 4.2628 | 6.6951 |

**Mesh-invariance evaluation.** We use $(u(\boldsymbol{X}, \boldsymbol{T}), u(\boldsymbol{X}, \boldsymbol{T'}))$ as the training set, and evaluate the model's performance of mesh-invariant inference ability on $\boldsymbol{X'}$, where $|\boldsymbol{X'}| = n_s'$. $\boldsymbol{X'}$ is a different mesh with $\boldsymbol{X} \subseteq \boldsymbol{X'}$. The visualization results of NS ($n_s = 4096, n_t' = 40$) are shown in Fig. 4. For a fixed $n_s'$, we randomly sampled different $\boldsymbol{X'}$ for 100 times, to get the mean errors and standard deviations (given in Appendix. B.5) of different spatial meshes. We can conclude from Table. 3 that *(1)* The errors on unseen meshes are larger than the errors on seen meshes, showing the overfitting effects. However, the errors on unseen meshes are acceptable, since they are even lower than other models' prediction error on seen meshes. *(2)* Larger $n_s'$ leads to higher prediction error because a large number of unseen points are likely to disturb the learned token mixing patterns. On the other hand, NS ($n_s = 1024, n_t' = 10$) implies that small spatial point numbers of training meshes ($n_s$) hinder model's generalizing ability on unseen meshes, due to excessive loss of spatial information.

### 4.3 ARCHITECTURE ANALYSIS

Two modules in NFS differ from FNO. The first is the interpolation layers at the beginning and the end of the architecture. The second is the extra Layer-Norm in the FNO layers, which can be applicable in NFS thanks to its fixed resampled equispaced points, but inapplicable to FNO for preserving its resolution-invariance. We aim to figure out what makes NFS outperform FNO.

**Effects of neighborhood sizes.** It is widely believed that modeling the long-range dependency among tokens brings improvements (Naseer et al., 2021; Tuli et al., 2021; Mao et al., 2021). By contrast, some local kernel methods demonstrate their superiority (Yang et al., 2019; Liu et al., 2021; Chu et al., 2021; Park & Kim, 2022). For this reason, we first conjecture that the large neighborhood sizes in the interpolation layer are conducive to predictive performance.

Besides, as demonstrated in Sec. 3.3, the patchwise embedding in Vision Mixers can be an analogy to the resampling and interpolating, so we further establish a patchwise FNO (PFNO), with patch size equaling to $4$ and $[4, 4]$ in 1-d and 2-d PDE problems, equivalent to each resampled points aggregating 4 and 16 points in spatial domains in 1-d and 2-d situations respectively. Layer-Norm is stacked in the FNO layers in PFNO, for a fair comparison. Results of Fig. 5 show that the long-range dependency may even compromise the performance, as larger mean neighborhood sizes often cause higher errors. However, no matter how large is the neighbor size, the NFS outperforms PFNO. More details are given

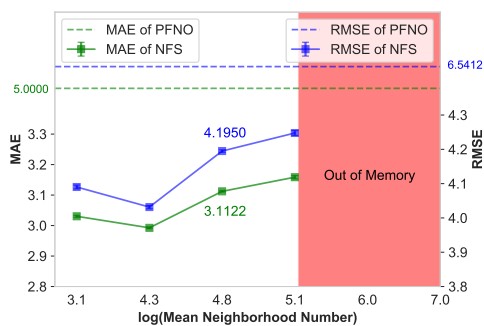

Figure 5: Effects of neighborhood sizes on NS ($r = 64, n'_t = 10, n'_t = 40$).

in Appendix. B.6. Therefore, we rule out the possibility of performance gains brought form large neighborhood sizes and suppose that proposed kernel interpolation layers are the key, and is superior to the simple patchwise embedding methods.

**Benefits from learned interpolation kernel.** Since the kernel interpolation is likely to hold the key to improvements, we investigate the performance gains brought from adaptively learned interpolation kernels over the predefined one (See Fig. 3). We use an inflexible Gaussian kernel $h(\boldsymbol{x}_j - \boldsymbol{x}_i) = \beta \exp(-(\boldsymbol{x}_j - \boldsymbol{x}_i - \boldsymbol{\mu})^T (\Gamma)^{-1} (\boldsymbol{x}_j - \boldsymbol{x}_i - \boldsymbol{\mu}))$ as a predefined one as discussed in Sec. 3.1, where $\Gamma = \mathrm{diag}(\gamma^{(1)}, \ldots, \gamma^{(d)})$, and $\boldsymbol{\mu} \in \mathbb{R}^d$, $\gamma^{(1)}, \ldots, \gamma^{(d)}, \beta \in \mathbb{R}^+$ are learnable parameters. By setting all the other modules and the interpolation neighborhood sizes as the same, we compare performance on different meshes of the two interpolation kernels in Table. 3 (Gaus + LN), where the adaptively learned kernels achieve better accuracy.

**Benefits from normalization layers.** Previous works demonstrated the normalization is necessary for network architecture, for fast convergence and stable training (Dong et al., 2021; Ba et al., 2016). A notable difference between NFS and FNO is that the Layer-Norm can be implemented in NFS's layers without disabling its discretization-invariance. The improvements brought from the normalization layers are given in Table. 3 (Flex + LN), where the performance gap is obvious on unseen meshes.

### 4.4 Non-equispaced Vision Mixers

Since NFS can be regarded as a combination of our interpolation layers with the revised FNO, our interpolation layers can also be implemented in the other Vision Mixers, so that these methods are equipped with the ability to handle non-equispaced data. Details are given in Appendix. B.7. We find that *(1)* From Table. B10 and Table. 1, the degeneration of performance is obvious in other Vision Mixers. In comparison, FNO as intermediate equispaced layers, truncates the high frequency in its channel mixing and retains the low frequency shared by both resampled and original signals, so the loss of accuracy in non-equispaced scenarios is tiny in our NFS; *(2)* Although the performance on unseen meshes is more stable in these non-equispaced Vision Mixers, the performance gap is still large, according to Table. B10 and Table. 3.

### 4.5 Complexity comparison

The discussed Fig. 2 shows time complexity of each method. Vision Mixers are the fastest, while graph spatio-temporal models are far much slower. NFS falls in between because the interpolation layer can be an analogy to a graph-message-passing layer (See Sec. 3.3), and the intermediate are equispaced token-channel mixing layers of Vision Mixers' structure. Memory usage shows storing each point's neighbors in the interpolation layer is very memory-consuming.

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

# A  METHODS

## A.1  NOTATION

| Symbol | Used for |
|---|---|
| $\boldsymbol{X}$ | A discretization of the domain $D$, which is used to train the model. |
| $\boldsymbol{X'}$ | A discretization of the domain $D$, which is used to evaluate the model. |
| $\boldsymbol{x}$ | Coordinate of spatial point in $D$. |
| $n_s$ | Number of spatial points of seen meshes for training, as $|\boldsymbol{X}| = n_s$. |
| $n_s'$ | Number of spatial points of unseen meshes for inference, as $|\boldsymbol{X'}| = n_s'$. |
| $n_t$ | Number of input timestamps as the number of input time-dependent PDE's initial states. |
| $n_t'$ | Number of output timestamps as the number of output time-dependent PDE's future states. |
| $a$ | Input function, where $a \in \mathcal{A}(D; \mathbb{R}^{d_a})$ means for $\boldsymbol{x} \in D$, $a(\boldsymbol{x}) \in \mathbb{R}^{d_a}$. In time-dependent PDEs, $a = u(\cdot, \boldsymbol{T})$, where $\boldsymbol{T} = \{t_i\}_{i=1}^{n_t}$. |
| $u$ | Target function for approximation, where $u \in \mathcal{U}(D; \mathbb{R}^{d_u})$ means for $\boldsymbol{x} \in D$, $u(\boldsymbol{x}) \in \mathbb{R}^{d_u}$. |
| $v$ | Representation function, where $v \in \mathcal{U}(D; \mathbb{R}^{d_v})$ means for $\boldsymbol{x} \in D$, $v(\boldsymbol{x}) \in \mathbb{R}^{d_v}$, which is a function obtained by a lifter which project $a$ into a higher dimensional space. |
| $\mathcal{G}_\theta$ | Approximation operator, where $\mathcal{G}_\theta(a) \approx u$. |
| $\mu$ | The probability measure for sampling Spatial points $\boldsymbol{x}$, which is supported on $D$. |
| $\mathcal{C}$ | Cost functional as the minimum optimization target. |
| $\mathcal{F}$ | Discrete Fourier transform for equispaced spatial points. |
| $\mathcal{F}^{-1}$ | Discrete Inverse Fourier transform for equispaced spatial points. |
| $P$ | Project operator, with $P(a)(\boldsymbol{x}) \in \mathbb{R}^{d_v}$. |
| $Q$ | Project operator, with $Q(v)(\boldsymbol{x}) \in \mathbb{R}^{d_u}$. |
| $v^{\mathrm{t}}$ | t-th iterative representation function after kernel operators' update. |
| $\mathcal{K}_\phi$ | Kernel integral operator mapping, which maps $a$ to a bounded linear operator, with parameter $\phi$. |
| $W$ | Linear transform on the $d_v$ dimension (channel) of $v(\boldsymbol{x}) \in d_v$. |
| $R_\phi$ | Fourier transform of a periodic kernel function, which is learnable parameters in a single iterative process in FNO. |
| ChannelMix | Channel-mixing operator as a linear transform in the dimensoion of channel ($d_v$). |
| TokenMix | Token-mixing operator as a linear transform in the dimensoion of spatial points ($n_s$). |
| $\tilde{\mathcal{F}}$ | Proposed non-equispaced Fourier transform, where $\tilde{\mathcal{F}} = (\mathcal{F} \circ \mathcal{H}(a))$. |
| $m_s$ | Spatial points' number on resampled equi-spaced points. |
| $\mathcal{H}$ | Interpolation operator to interpolate the non-equispaced spatial points on equispaced spatial grids. |
| $\tau$ | Parameter in Gaussian interpolation kernels controlling smoothness of the kernel. |
| $\boldsymbol{\mu}$ | Parameter in Gaussian interpolation kernels, as the mean of Gaussian kernels. |
| $\mathcal{H}_\eta$ | Interpolation operator mapping, where $\mathcal{H}_\eta(a)$ is an interpolation operator used to map signals on the non-equispaced spatial points to equispaced spatial grids. |
| $\mathcal{H}'_\zeta$ | Interpolation operator mapping, where $\mathcal{H}'_\zeta(a)$ is a interpolation operator used to map signals on the equispaced spatial points to non-equispaced spatial grids. |
| $\mathcal{N}(\boldsymbol{x})$ | Neighborhood of spatial point $\boldsymbol{x}$. |

Table A1: Glossary of Notations used in this paper.

## A.2 Graph construction

**Neighborhood construction.** Instead of using K-nearest neighborhood method, the neighborhood system in the interpolation layer is constructed by $\epsilon$-ball, because in equispace scenarios, there will be multiple points as K-th nearest neighbor at the same time. For point $\boldsymbol{x}$, its neighbor is defined according to

$$\begin{cases} d(\boldsymbol{x}, \boldsymbol{x}_i) \leq \epsilon & \boldsymbol{x}_i \in \mathcal{N}(\boldsymbol{x}); \\ d(\boldsymbol{x}, \boldsymbol{x}_i) > \epsilon & \boldsymbol{x}_i \notin \mathcal{N}(\boldsymbol{x}). \end{cases} \tag{10}$$

For given $c$ defined in Sec. 3.2, we can restrict $\epsilon$ so that $\mathbb{E}_{x \sim \mu}[|\mathcal{N}(\boldsymbol{x})|] < c \log(n_s)$.

## A.3 Proof of Theorem 3.1.

Our proof is mostly based on Chen & Chen (1995) and Kovachki et al. (2021). For notation simplicity, in the proof, we directly write $\mathcal{H}_\eta(a)$ as $\mathcal{H}_\eta$ as the linear operator.

**Lemma A1.** *Let $\mathcal{X}$ be a Banach space, and $\mathcal{U} \subseteq \mathcal{X}$ a compact set, and $\mathcal{K} \subset \mathcal{X}$ a dense set. Then, for any $\epsilon > 0$, there exists a number $n \in \mathbb{N}$, and a series of continuous, linear functionals $G_1, G_2, \ldots, G_n \in C(\mathcal{U}; \mathbb{R})$, and elements $\varphi_1, \ldots, \varphi_n \in \mathcal{K}$, such that*

$$\sup_{u \in \mathcal{U}} ||v - \sum_{j=1}^n G_j(v)\varphi_j||_\mathcal{X} \leq \epsilon \tag{11}$$

The proof is given in **Lemma 7.** in Kovachki et al. (2021), and **Theorem 3.** and **4.** for reference .

**Theorem A2.** *Let $D \subseteq \mathbb{R}^d$ be compact domain. Let $\mathcal{U}$ be a separable Banach space of real-valued functions on $D$, such that $C(D, \mathbb{R}) \subseteq \mathcal{U}$ is dense. Suppose $\mathcal{U} = L^p(D; \mathbb{R})$ for any $1 < p < \infty$. $\nu$ is a probability measure supported on $\mathcal{U}$ and assume that, $\mathbb{E}_{v \sim \nu}||v||_\mathcal{U} < \infty$ for any $v \in \mathcal{U}$. $\mu$ is a probabilistic measure supported on $D$, which defines the inner product of Hilbert space $\mathcal{U}$ as $< f, g >_\mathcal{U} = \int_D f(\boldsymbol{x})g(\boldsymbol{x})d\mu(\boldsymbol{x})$. Then, there exists a neural network $h_\eta : \mathbb{R}^d \times \mathbb{R}^d \to \mathbb{R}$ whose activation functions are of the Tauber-Wiener class, such that*

$$||v - \mathcal{H}(v)||_\mathcal{U} \leq \epsilon,$$

*where $\mathcal{H}(v)(\boldsymbol{x}) = \int_D h_\eta(\boldsymbol{x}, \boldsymbol{y})v(\boldsymbol{y})d\mu(\boldsymbol{y})$.*

*Proof.* Since $\mathcal{U}$ is a Polish space, we can find a compact set $\mathcal{K}$, such that $\nu(\mathcal{U} \setminus \mathcal{K}) \leq \epsilon$. Therefore, **Lemma A1** can be applied, to find a number $n \in \mathbb{N}$, a series of continuous linear functionals $G_j \in C(\mathcal{U}; \mathbb{R})$ and functions $\varphi_j \in C(D; \mathbb{R})$ such that

$$\sup_{v \in \mathcal{K}} ||v - \sum_{j=1}^n G_j(v)\varphi_j||_\mathcal{U} \leq \epsilon.$$

Denote $\hat{\mathcal{H}}_n(v) = \sum_{j=1}^n G_j(v)\varphi_j$, and let $1 < q < \infty$ be the Hölder conjugate of $p$. Since $\mathcal{U} = L^p(D; \mathbb{R})$, by Reisz Representation Theorem, there exists functions $g_j \in L^q(D; \mathbb{R})$, such that $G_j(v) = \int_D v(\boldsymbol{x})g_j(\boldsymbol{x})d\mu(\boldsymbol{x})$ for $j = 1, \ldots, n$ and $v \in L^p(D; \mathbb{R})$. By density of $C(D; \mathbb{R})$ in $L^q(D; \mathbb{R})$, we can find functions $\psi_1, \ldots, \psi_n \in C(D; \mathbb{R})$, such that

$$\sup_{j \in \{1, \ldots, n\}} ||\psi_j - g_j||_{L^q(D; \mathbb{R})} \leq \epsilon/n.$$

Then, we define $\tilde{\mathcal{H}}_n : L^p(D; \mathbb{R}) \to C(D; \mathbb{R})$ by

$$\tilde{\mathcal{H}}_n(v) = \sum_{j=1}^n \int_D \psi_j(\boldsymbol{y})v(\boldsymbol{y})d\mu(\boldsymbol{y})\varphi_j(\boldsymbol{x}).$$

For the universal approximation (density) (Hornik et al., 1989) of neural networks, we can find a Multi-layer Feedforward network $h_\eta : \mathbb{R}^d \times \mathbb{R}^d \to \mathbb{R}$ whose activation functions are of the Tauber-Wiener class, such that

$$\sup_{\boldsymbol{x}, \boldsymbol{y} \in D} |h_\eta(\boldsymbol{x}, \boldsymbol{y}) - \sum_{j=1}^n \psi_j(\boldsymbol{y})\varphi_j(\boldsymbol{x})| \leq \epsilon.$$

Let $\mathcal{H}_\eta(\boldsymbol{x}) = \int_D h_\eta(\boldsymbol{x}, \boldsymbol{y})v(\boldsymbol{y})d\mu(\boldsymbol{y})$. Then, there exists a constant $C_1 > 0$, such that

$$||\hat{\mathcal{H}}_n(v) - \mathcal{H}(v)||_{L^p(D;\mathbb{R})} \le C_1(||\hat{\mathcal{H}}_n(v) - \tilde{\mathcal{H}}_n(v)||_{L^p(D;\mathbb{R})} + ||\tilde{\mathcal{H}}_n(v) - \mathcal{H}(v)||_{L^p(D;\mathbb{R})}).$$

For the first term, there is a constant $C_2 > 0$, such that

$$||\hat{\mathcal{H}}_n(v) - \tilde{\mathcal{H}}_n(v)||_{L^p(D;\mathbb{R})} \le C_2 \sum_{j=1}^n ||\int_D v(\boldsymbol{y})(g_j(\boldsymbol{y}) - \psi_j(\boldsymbol{y}))d\mu(\boldsymbol{y})\varphi_j||_{L^p(D;\mathbb{R})}$$

$$\le C_2 \sum_{j=1}^n ||v(\boldsymbol{y})||_{L^p(D;\mathbb{R})}||g_j(\boldsymbol{y}) - \psi_j(\boldsymbol{y})||_{L^q(D;\mathbb{R})}||\varphi_j||_{L^p(D;\mathbb{R})}$$

$$\le C_3\epsilon||v(\boldsymbol{y})||_{L^p(D;\mathbb{R})},$$

for some $C_3 > 0$. And for the second term,

$$||\tilde{\mathcal{H}}_n(v) - \mathcal{H}(v)||_{L^p(D;\mathbb{R})} = ||\int_D v(\boldsymbol{y})(\sum_{j=1}^n \psi_j(\boldsymbol{y})\varphi_j(\cdot) - h_\eta(\cdot, \boldsymbol{y}))d\mu(\boldsymbol{y})||_{L^p(D;\mathbb{R})}$$

$$\le |D|\epsilon||v||_{L^p(D;\mathbb{R})},$$

Therefore, there is a constant $C > 0$, such that

$$\int_{\mathcal{U}} ||\hat{\mathcal{H}}_n(v) - \tilde{\mathcal{H}}_n(v)||_{\mathcal{U}}d\nu(v) \le \epsilon C\mathbb{E}_{v\sim\nu}||v||_{\mathcal{U}}$$

. Because of the assumption that $\mathbb{E}_{v\sim\nu}||v||_{\mathcal{U}} < \infty$, and $\epsilon$ is arbitrary, then

$$||v - \mathcal{H}(v)||_{\mathcal{U}} \le ||v - \hat{\mathcal{H}}_n(v)||_{\mathcal{U}} + ||\hat{\mathcal{H}}_n(v) - \mathcal{H}(v)||_{\mathcal{U}},$$

the proof is complete. $\square$

**Corollary A3.** *Define $\mathcal{H}_\eta(v) = \int_D h_\eta(\boldsymbol{x} - \boldsymbol{y}, \boldsymbol{x}, a(\boldsymbol{y}))v(\boldsymbol{y})d\mu(\boldsymbol{y})$, the interpolation operator can also approximate $v$ to any precision $\epsilon$.*

*Proof.* We use a one-layer neural network $h_\eta : D \times D \to \mathbb{R}$ as an example, which is defined as $h_\eta(\boldsymbol{x}, \boldsymbol{y}, a(\boldsymbol{y}) = \sigma(\sum_{i=1}^d w_{x,i}x^{(i)} + w_{y,i}y^{(i)} + b)$. We can rewrite it as

$$h_\eta = \sigma(\sum_{i=1}^d w_{x,i}(x^{(i)} - y^{(i)}) + (w_{y,i} + w_{x,i})y^{(i)} + \sum_{j=1}^{d_a} w_{a,j}a^{(j)}(\boldsymbol{y}) + b),$$

where $w_{a,j} = 0$. $\square$

**Corollary A4.** *The **Theorem A2** and **Corollary A3** can be extended for $v : D \to \mathbb{R}^{d_v}$, where $d_v > 1$.*

*Proof.* As $v = (v^{(1)}, v^{(2)}, \dots, v^{(d_v)})$, for each $v^{(j)}$, a single neural network can be used for approximation. Moreover, in implementation, we make $h_\eta$ fully-connected, to improve the expressivity. $\square$

**Remark.** *As $\sum_{\boldsymbol{x}_i \in \boldsymbol{X}} v(\boldsymbol{x}_i)h_\eta(\boldsymbol{x} - \boldsymbol{x}_i, \boldsymbol{x}_i, a(\boldsymbol{x}_i))$ is the unbiased estimation of $\mathbb{E}_{\boldsymbol{y}\sim\mu}(h_\eta(\boldsymbol{x}, \boldsymbol{y})v(\boldsymbol{y}))$, we use the Equation. (8) for the approximation.*

# B  EXPERIMENTS

## B.1  BENCHMARK METHOD DESCRIPTION

**Vision Mixers.**   We provide a framework for vision mixers as PDE solvers, including VIT, MLP-MIXER, FNET, GFN, FNO, PFNO and our NFS. The intermediate architecture of mixing layers is shown in Fig. B1. The code of our framework will be released soon. And the resampling and back-sampling methods are stacked before 'Equispaced Input' and 'Equispaced Output'. In this way, the description of the Vision Mixers included in our framework can be described by different modules, as shown in Table. B1. All the trials on Vision Mixers set *embedding size* as 32, *batch size* as 4, *layer number* of the intermediate equispaced mixing layers as 2. In FNO and PFNO, the truncated $K_{\max}$ is set as 16. The *patch size* of Vision Mixers with patchwise embedding are set as $[4, 2]$ in 1-d PDEs and $[4, 4, 2]$ in 2-d PDEs. The interpolation layers in NFS are composed of one layer of feed-forward network whose perceptron unit is equal to $4\times$ *embedding size* of the model.

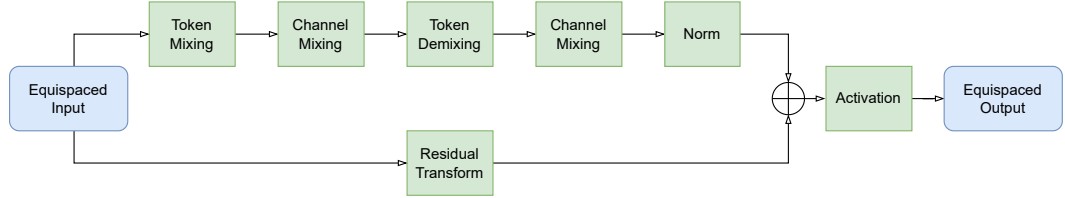

Figure B1: The architecture of Vision Mixers.

Table B1: Description of Vison Mixers in the unifying framework module by module.

| Modules | VIT | MLPMIXER | FNET | GFN | FNO | PFNO | NFS |
|---|---|---|---|---|---|---|---|
| Resampling | Patchwise Embedding | Patchwise Embedding | Patchwise Embedding | Patchwise Embedding | Identity | Patchwise Embedding | Interpolation |
| Token Mixing | Attention | MLP | Fourier | Fourier | Fourier | Fourier | Fourier |
| Channel Mixing | Linear | Linear | Linear | Elementwise Product | Low Frequency MatMultiply | Low Frequency MatMultiply | Low Frequency MatMultiply |
| Token Demixing | Identity | Identity | Identity | Inverse Fourier | Inverse Fourier | Inverse Fourier | Inverse Fourier |
| Channel Mixing | Identity | Identity | Identity | Linear | Identity | Identity | Identity |
| Normalization | LayerNorm | LayerNorm | Complex LayerNorm | LayerNorm | Identity | LayerNorm | LayerNorm |
| Residual | Identity | Identity | Identity | Identity | 1x1 Conv | 1x1 Conv | 1x1 Conv |
| Activation | Gelu | Gelu | Complex Gelu | Gelu | Gelu | Gelu | Gelu |
| Back Sampling | Linear+ Rearrange | Linear+ Rearrange | Linear+ Rearrange | Linear+ Rearrange | Identity | Linear+ Rearrange | Interpolation |

**DeepONet Variants.** Since vanilla DeepONet uses MLP as Branch Net, it cannot be implemented in such a high-resolution dataset, because for a resolution like the trial (NS $n_s = 4096, n_t = 10, n_t' = 10$), DeepONet assigns each data point a weight parameter in a single MLP, leading the MLP's parameter number reaches $O(n_s^2 n_t^2) \approx 40960^2$ in a single Branch Net, which is infeasible in practice. In the original paper, the spatial point's number in the experiments is set as 40, far less than in the recent Neural Operator's evaluation protocol.

One feasible alternative is to use other architecture to replace the original MLP, thus allowing DeepONet to handle high-resolution data. For example, CNN and Vit. Therefore, we here conduct further experiments on the three equations in the context, to evaluate DeepONet-U (using UNet as the Branch Net) and DeepONet-V (using Vit as the Branch Net) as two variants of vanilla DeepONet for comparison. Note that the architecture of variants of DeepONet are all limited to equispaced data.

**Graph Spatio-Temporal Models.** The evaluated graph spatio-temporal neural networks are based on recurrent neural networks for dynamics modeling, where the spatial dependency is modeled by graph neural networks. The spatial and temporal modules for AGCRN, DCRNN and GCGRU are shown in Table. B2. MPPDE used different architecture, with the pushforward trick used for taining, with *rolling* equaling 1 and *time window* equaling to 10 . All the trials on these graph spatio-tempral models set *embedding size* as 64, except MPPDE as 128. *Batch size* is set as 4. When the graph convolution needs multi-hop message-passing, we set the hop as 2. For MPPDE, the layer number of GNNs is 6. The *embedding dimension* in AGCRN is set as 2.

Table B2: Description of different graph spatio-temporal models

| Methods | Spatial module | Temporal module |
|---|---|---|
| GCGRU Seo et al. (2016) | Cheb Conv Defferrard et al. (2017) | GRU |
| DCRNN Li et al. (2018) | Diff Conv Atwood & Towsley (2016) | GRU |
| AGCRN Bai et al. (2020) | Node Similarity Bai et al. (2020) | GRU |

## B.2 DATA GENERATION

**Burgers' Equation.** The initial condition $u_0(x)$ is generated according to $u_0 \sim N(0.625(-\Delta + 25I)^{-2})$ with periodic boundary conditions. $\nu$ is set as 0.01. $x \in [0,1]$ and $t \in [0,1]$. The spatial resolution is 1024, and time resolution is 200. The dataset generation follows FNO's protocol, which can be downloaded from its source code on official Github.

**KdV Equation.** The equation is written as

$$\partial_t u(x,t) + 3\partial_x u^2(x,t) + \partial_x^3 u(x,t) = 0, \tag{12}$$

where $x \in [0,1]$. The initial condition $u_0(x)$ is calculated as

$$u(x,0) = \sum_{i=1}^{K} 0.5 c_i \cos(0.5\sqrt{c_i + b_i}x - a_i)$$

where $c_i \sim N(0, \sigma_i)$, and $a_i, b_i > 0$. The spatial resolution is 1024. The dataset is generated by `scipy` package, with `fftpack.diff` used as pesudo-differential method and `odeint` used as forward Euler method.

**Darcy Flow.** The equation is written as

$$\begin{aligned}
-\nabla(a(\boldsymbol{x})\nabla u(\boldsymbol{x})) = f(\boldsymbol{x}) && \boldsymbol{x} \in (0,1)^2 \\
u(\boldsymbol{x}) = 0 && \boldsymbol{x} \in \partial[0,1]^2
\end{aligned} \tag{13}$$

The original resolution is $256 \times 256$. $a(\boldsymbol{x})$ is generated by Gaussian random field, and we directly establish the operator to learn the mapping of $a$ to $u$.

**NS Equation.** Our generation of NS Equation is based on FNO's Appendix. A.3.3, with the forcing is kept fixed. The original spatial resolution is $128 \times 128$, and time resolution is 200.

## B.3 COMPLETE RESULTS ON MODEL COMPARISON

Here we give complete results on the four Equations. Table. B3 give the performance comparison on Darcy flow of both equispaced and non-equispaced scenarios. Table. B4 and B5 gives performance comparison in equispaced scenarios on the other three time-dependent problems. Table. B6 and B7 gives performance comparison in non-equispaced scenarios on the other three time-dependent problems. In all the tasks except `Darcy Flow`, the depth of layer is set as 2, and $k_{\max} = 16$ in both NFS and FNO. However, we find in `Darcy Flow`, $k_{\max}$ should be set much larger, or the loss will not decrease. In the reported results, $k_{\max} = 32, 64, 128$ in `Darcy Flow`.

Table B3: Performance comparison on Darcy Flow.

| | MAE ($\times 10^{-3}$) | RMSE($\times 10^{-3}$) | MAE($\times 10^{-3}$) | RMSE($\times 10^{-3}$) | MAE($\times 10^{-3}$) | RMSE($\times 10^{-3}$) |
|---|---|---|---|---|---|---|
| | Darcy Flow ($r = 64$) | | Darcy Flow ($r = 128$) | | Darcy Flow ($r = 256$) | |
| VIT | $0.5073_{\pm 0.0411}$ | $0.8468_{\pm 0.0432}$ | $0.9865_{\pm 0.0002}$ | $1.6195_{\pm 0.0007}$ | $1.1078_{\pm 0.0021}$ | $1.8444_{\pm 0.0023}$ |
| MLPMIXER | $0.4970_{\pm 0.0021}$ | $0.8228_{\pm 0.0034}$ | $0.8909_{\pm 0.0099}$ | $1.4221_{\pm 0.0118}$ | $0.9125_{\pm 0.0024}$ | $1.6459_{\pm 0.0032}$ |
| GFN | $0.4739_{\pm 0.0016}$ | $0.8345_{\pm 0.0019}$ | $0.8659_{\pm 0.0046}$ | $1.4237_{\pm 0.0071}$ | $0.9618_{\pm 0.0124}$ | $1.6139_{\pm 0.0128}$ |
| FNO | $0.4289_{\pm 0.0051}$ | $0.7740_{\pm 0.0046}$ | $0.7086_{\pm 0.0045}$ | $0.1324_{\pm 0.0019}$ | $0.9075_{\pm 0.0051}$ | $1.4940_{\pm 0.0046}$ |
| NFS | $\mathbf{0.1497}_{\pm 0.0005}$ | $\mathbf{0.1962}_{\pm 0.0007}$ | $\mathbf{0.2254}_{\pm 0.0007}$ | $\mathbf{0.7245}_{\pm 0.0009}$ | $\mathbf{0.4216}_{\pm 0.0033}$ | $\mathbf{0.8578}_{\pm 0.0041}$ |
| | Darcy Flow ($n_s = 1024$) | | Darcy Flow ($n_s = 4096$) | | Darcy Flow ($n_s = 16384$) | |
| DCRNN | $1.8146_{\pm 0.0060}$ | $2.6352_{\pm 0.0029}$ | $1.7629_{\pm 0.0003}$ | $2.5760_{\pm 0.0001}$ | OOM | OOM |
| AGCRN | $1.6938_{\pm 0.0001}$ | $2.4440_{\pm 0.0001}$ | $1.7336_{\pm 0.0001}$ | $2.4167_{\pm 0.0001}$ | OOM | OOM |
| GCGRU | $1.7633_{\pm 0.0001}$ | $2.5696_{\pm 0.0001}$ | $1.7403_{\pm 0.0001}$ | $2.5363_{\pm 0.0001}$ | OOM | OOM |
| MPPDE | $0.6673_{\pm 0.0009}$ | $0.9290_{\pm 0.0012}$ | $0.5608_{\pm 0.0053}$ | $0.8424_{\pm 0.0051}$ | $0.6384_{\pm 0.0005}$ | $0.8748_{\pm 0.0005}$ |
| NFS | $\mathbf{0.1727}_{\pm 0.0047}$ | $\mathbf{0.2311}_{\pm 0.0066}$ | $\mathbf{0.1430}_{\pm 0.0007}$ | $\mathbf{0.1914}_{\pm 0.0014}$ | $\mathbf{0.2379}_{\pm 0.0007}$ | $\mathbf{0.3489}_{\pm 0.0009}$ |

NFS fails to model the non-equispaced Burgers' Equation when $n_t$ is set as 1, in which the performance is far from it can achieve in equispaced scenarios. Such problem will be our future work.

Table B4: Performance comparison with Vision Mixer benchmarks on different equations $(n_t = 1)$. Validation loss on `Burgers'` $(n_t = 1)$ of VIT, GFN, and FNO does not converge. The results show that the early-stopping occurs in the begining of training.

| Vision Mixers | MAE $(\times 10^{-3})$ | RMSE $(\times 10^{-3})$ | MAE $(\times 10^{-3})$ | RMSE $(\times 10^{-3})$ | MAE $(\times 10^{-3})$ | RMSE $(\times 10^{-3})$ |
|---|---|---|---|---|---|---|
| | `Burgers'` $(r=512, n_t'=10)$ | | `Burgers'` $(r=512, n_t'=40)$ | | `Burgers'` $(r=1024, n_t'=20)$ | |
| VIT | $201.6539_{\pm 0.5284}$ | $231.9138_{\pm 0.8403}$ | $183.6696_{\pm 0.3015}$ | $210.6237_{\pm 0.6767}$ | $195.5858_{\pm 0.7706}$ | $224.4712_{\pm 1.2472}$ |
| MLPMIXER | $201.6547_{\pm 0.0671}$ | $231.9163_{\pm 0.0263}$ | $183.6535_{\pm 0.0599}$ | $210.6160_{\pm 0.0305}$ | $195.5960_{\pm 0.0240}$ | $224.4791_{\pm 0.0132}$ |
| GFN | $201.6557_{\pm 0.9513}$ | $231.9122_{\pm 0.9535}$ | $183.6674_{\pm 0.4893}$ | $210.6165_{\pm 0.4831}$ | $195.5918_{\pm 0.0471}$ | $224.4736_{\pm 0.0681}$ |
| FNO | $201.6527_{\pm 1.1415}$ | $231.9119_{\pm 1.6747}$ | $183.6696_{\pm 0.3015}$ | $210.6299_{\pm 0.4983}$ | $195.5902_{\pm 0.9304}$ | $224.4723_{\pm 0.9230}$ |
| NFS | $\mathbf{0.1806}_{\pm 0.0005}$ | $\mathbf{0.2669}_{\pm 0.0010}$ | $\mathbf{0.3570}_{\pm 0.0008}$ | $\mathbf{0.5340}_{\pm 0.0009}$ | $\mathbf{0.4344}_{\pm 0.0014}$ | $\mathbf{0.6092}_{\pm 0.0017}$ |
| | KdV $(r=512, n_t'=10)$ | | KdV $(r=512, n_t'=40)$ | | KdV $(r=1024, n_t'=20)$ | |
| VIT | $0.2808_{\pm 0.0006}$ | $0.3938_{\pm 0.0009}$ | $0.3428_{\pm 0.0012}$ | $0.6832_{\pm 0.0016}$ | $0.3066_{\pm 0.0003}$ | $0.5461_{\pm 0.0003}$ |
| MLPMIXER | $0.2732_{\pm 0.0054}$ | $0.4259_{\pm 0.0088}$ | $\mathbf{0.3336}_{\pm 0.0045}$ | $\mathbf{0.5923}_{\pm 0.0081}$ | $0.2872_{\pm 0.0005}$ | $0.5235_{\pm 0.0006}$ |
| GFN | $0.2587_{\pm 0.0032}$ | $0.3490_{\pm 0.0056}$ | $0.3086_{\pm 0.0223}$ | $0.5952_{\pm 0.0338}$ | $0.2011_{\pm 0.0074}$ | $0.3464_{\pm 0.0063}$ |
| FNO | $0.2619_{\pm 0.0069}$ | $0.3849_{\pm 0.0107}$ | $0.5608_{\pm 0.0053}$ | $0.8424_{\pm 0.0051}$ | $0.3925_{\pm 0.0079}$ | $0.4623_{\pm 0.0087}$ |
| NFS | $\mathbf{0.2514}_{\pm 0.0008}$ | $\mathbf{0.3776}_{\pm 0.00011}$ | $0.4522_{\pm 0.0013}$ | $0.6290_{\pm 0.0022}$ | $\mathbf{0.2254}_{\pm 0.0007}$ | $\mathbf{0.0745}_{\pm 0.0010}$ |
| | NS $(r=64, n_t'=10)$ | | NS $(r=64, n_t'=40)$ | | NS $(r=128, n_t'=20)$ | |
| VIT | $9.3797_{\pm 0.0421}$ | $12.9291_{\pm 0.0703}$ | $22.8565_{\pm 0.0935}$ | $29.1130_{\pm 0.1428}$ | $15.7398_{\pm 0.0757}$ | $20.6927_{\pm 0.0664}$ |
| MLPMIXER | $7.5246_{\pm 0.0080}$ | $10.4762_{\pm 0.0096}$ | $15.8632_{\pm 0.0375}$ | $20.1522_{\pm 0.0604}$ | $14.9360_{\pm 0.0305}$ | $19.3268_{\pm 0.0635}$ |
| GFN | $3.5524_{\pm 0.0057}$ | $4.7071_{\pm 0.0088}$ | $10.2250_{\pm 0.0331}$ | $13.0451_{\pm 0.0704}$ | $6.3976_{\pm 0.00345}$ | $8.2685_{\pm 0.297}$ |
| FNO | $3.3425_{\pm 0.0007}$ | $5.2566_{\pm 0.0008}$ | $8.9857_{\pm 0.0010}$ | $14.0171_{\pm 0.0023}$ | $4.4627_{\pm 0.0004}$ | $6.3047_{\pm 0.0004}$ |
| NFS | $\mathbf{1.7425}_{\pm 0.0017}$ | $\mathbf{2.2847}_{\pm 0.0022}$ | $\mathbf{4.7882}_{\pm 0.0066}$ | $\mathbf{6.1508}_{\pm 0.0042}$ | $\mathbf{2.6988}_{\pm 0.0005}$ | $\mathbf{3.5121}_{\pm 0.0006}$ |

Table B5: Performance comparison with Vision Mixer benchmarks on different equations $(n_t = 10)$.

| Vision Mixers | MAE $(\times 10^{-3})$ | RMSE $(\times 10^{-3})$ | MAE $(\times 10^{-3})$ | RMSE $(\times 10^{-3})$ | MAE $(\times 10^{-3})$ | RMSE $(\times 10^{-3})$ |
|---|---|---|---|---|---|---|
| | `Burgers'` $(r=512, n_t'=10)$ | | `Burgers'` $(r=512, n_t'=40)$ | | `Burgers'` $(r=1024, n_t'=20)$ | |
| VIT | $0.5042_{\pm 0.0114}$ | $0.7667_{\pm 0.0225}$ | $2.4269_{\pm 0.0288}$ | $3.7728_{\pm 0.0431}$ | $1.5327_{\pm 0.0314}$ | $2.4093_{\pm 0.0408}$ |
| MLPMIXER | $0.1973_{\pm 0.0070}$ | $0.2600_{\pm 0.0097}$ | $0.4210_{\pm 0.0084}$ | $0.5844_{\pm 0.0101}$ | $0.3303_{\pm 0.0077}$ | $0.4473_{\pm 0.0086}$ |
| GFN | $0.2383_{\pm 0.0082}$ | $0.3066_{\pm 0.0114}$ | $0.4187_{\pm 0.0079}$ | $0.5407_{\pm 0.0090}$ | $0.3500_{\pm 0.0062}$ | $0.4489_{\pm 0.0081}$ |
| FNO | $0.0978_{\pm 0.0019}$ | $\mathbf{0.1287}_{\pm 0.0023}$ | $0.1815_{\pm 0.0009}$ | $0.2410_{\pm 0.0011}$ | $\mathbf{0.1430}_{\pm 0.0009}$ | $\mathbf{0.1871}_{\pm 0.0010}$ |
| NFS | $\mathbf{0.0958}_{\pm 0.0015}$ | $0.1347_{\pm 0.0022}$ | $\mathbf{0.1708}_{\pm 0.0006}$ | $\mathbf{0.2351}_{\pm 0.0009}$ | $0.1474_{\pm 0.0026}$ | $0.1957_{\pm 0.0034}$ |
| | KdV $(r=512, n_t'=10)$ | | KdV $(r=512, n_t'=40)$ | | KdV $(r=1024, n_t'=20)$ | |
| VIT | $0.2066_{\pm 0.0027}$ | $0.3525_{\pm 0.0049}$ | $0.2376_{\pm 0.0022}$ | $0.5521_{\pm 0.0036}$ | $0.1897_{\pm 0.0003}$ | $0.3725_{\pm 0.0009}$ |
| MLPMIXER | $0.2152_{\pm 0.0023}$ | $0.3686_{\pm 0.0039}$ | $\mathbf{0.2497}_{\pm 0.0017}$ | $0.5400_{\pm 0.0029}$ | $0.2062_{\pm 0.0007}$ | $0.4429_{\pm 0.0012}$ |
| GFN | $0.1530_{\pm 0.0004}$ | $0.2607_{\pm 0.0006}$ | $0.2691_{\pm 0.0007}$ | $0.5451_{\pm 0.0014}$ | $0.1984_{\pm 0.0002}$ | $0.3869_{\pm 0.0003}$ |
| FNO | $0.3230_{\pm 0.0035}$ | $1.1105_{\pm 0.0061}$ | $0.9605_{\pm 0.0024}$ | $2.7500_{\pm 0.0055}$ | $0.5929_{\pm 0.0020}$ | $1.6473_{\pm 0.0033}$ |
| NFS | $\mathbf{0.0678}_{\pm 0.0002}$ | $\mathbf{0.1214}_{\pm 0.0003}$ | $0.2709_{\pm 0.0009}$ | $\mathbf{0.5122}_{\pm 0.0013}$ | $\mathbf{0.1576}_{\pm 0.0003}$ | $\mathbf{0.3114}_{\pm 0.0005}$ |
| | NS $(r=64, n_t'=10)$ | | NS $(r=64, n_t'=40)$ | | NS $(r=128, n_t'=20)$ | |
| VIT | $3.9609_{\pm 0.0101}$ | $6.0575_{\pm 0.0250}$ | $12.3433_{\pm 0.0342}$ | $16.5238_{\pm 0.0415}$ | $9.3010_{\pm 0.0234}$ | $14.0027_{\pm 0.0380}$ |
| MLPMIXER | $3.1530_{\pm 0.0049}$ | $4.4339_{\pm 0.0067}$ | $7.9291_{\pm 0.0038}$ | $10.4149_{\pm 0.0066}$ | $7.7410_{\pm 0.0037}$ | $10.1934_{\pm 0.0082}$ |
| GFN | $1.7396_{\pm 0.0016}$ | $2.3551_{\pm 0.0028}$ | $5.4464_{\pm 0.0023}$ | $7.2130_{\pm 0.0032}$ | $3.1261_{\pm 0.0026}$ | $4.1691_{\pm 0.0047}$ |
| FNO | $2.4076_{\pm 0.0017}$ | $3.2861_{\pm 0.0024}$ | $7.6979_{\pm 0.0035}$ | $10.6401_{\pm 0.0056}$ | $3.7001_{\pm 0.0034}$ | $5.0047_{\pm 0.0072}$ |
| NFS | $\mathbf{0.8636}_{\pm 0.0008}$ | $\mathbf{1.2264}_{\pm 0.0011}$ | $\mathbf{3.1122}_{\pm 0.0020}$ | $\mathbf{4.1950}_{\pm 0.0037}$ | $\mathbf{1.8406}_{\pm 0.0003}$ | $\mathbf{2.5620}_{\pm 0.0005}$ |

Table B6: Performance comparison with graph spatio-temporal benchmarks ($n_t = 1$).

| Graph Spatio-Temporal Models | MAE ($\times 10^{-3}$) | RMSE($\times 10^{-3}$) | MAE($\times 10^{-3}$) | RMSE($\times 10^{-3}$) | MAE($\times 10^{-3}$) | RMSE($\times 10^{-3}$) |
|---|---|---|---|---|---|---|
| | Burgers' ($n_s = 512, n'_t = 10$) | | Burgers' ($n_s = 256, n'_t = 20$) | | Burgers' ($n_s = 512, n'_t = 40$) | |
| DCRNN | 277.8393±0.0082 | 346.1716±0.0088 | 292.1712±0.0280 | 368.1883±0.0204 | 298.4096±0.0137 | 373.0938±0.0186 |
| AGCRN | 289.9780±0.0001 | 360.9834±0.0001 | 272.6697±0.3404 | 340.1351±0.5435 | 305.4976±0.2120 | 376.0804±0.2385 |
| GCGRU | 288.4507±0.0246 | 361.1175±0.0512 | 294.9075±0.0005 | 367.4703±0.0004 | 291.0365±0.0265 | 365.1668±0.0827 |
| MPPDE | 24.4997±0.0014 | 34.5123±0.0017 | 25.4357±0.0002 | 31.7015±0.0002 | 25.3311±0.0004 | 33.7808±0.0005 |
| NFS | **16.1860**±0.0016 | **28.1504**±0.0021 | **21.1634**±0.0018 | **33.8976**±0.0018 | **26.0818**±0.0001 | **44.7962**±0.0003 |
| | KdV ($n_s = 512, n'_t = 10$) | | KdV ($n_s = 256, n'_t = 20$) | | KdV ($r = 512, n'_t = 40$) | |
| DCRNN | 1.6855±0.0001 | 3.0875±0.0001 | 3.1267±0.0001 | 4.8662±0.0001 | 5.7387±0.0001 | 8.3752±0.0001 |
| AGCRN | 4.0753±0.0001 | 6.8943±0.0001 | 5.4107±0.0001 | 9.2333±0.0001 | 8.4438±0.0001 | 13.8677±0.0001 |
| GCGRU | 1.6554±0.0001 | 2.6839±0.0001 | 3.0677±0.0001 | 4.6557±0.0001 | 5.8745±0.0001 | 9.4528±0.0001 |
| MPPDE | 1.5452±0.0001 | 2.6774±0.0001 | 2.9929±0.0007 | 5.4582±0.0010 | 3.0101±0.0001 | 4.9946±0.0001 |
| NFS | **0.0816**±0.0012 | **0.1512**±0.0022 | **0.1576**±0.0007 | **0.3114**±0.0018 | **0.3210**±0.0021 | **0.6873**±0.0049 |
| | NS ($n_s = 4096, n'_t = 10$) | | NS ($n_s = 1024, n'_t = 20$) | | NS ($n_s = 4096, n'_t = 40$) | |
| DCRNN | 30.6756±0.0001 | 41.7815±0.0001 | 52.1290±0.0138 | 69.7019±0.0032 | 88.3382±0.0864 | 119.5021±0.0055 |
| AGCRN | OOM | OOM | 59.9393±0.0001 | 79.0434±0.0001 | OOM | OOM |
| GCGRU | 28.8537±0.0019 | 40.1215±0.0008 | 49.9352±0.0028 | 67.5623±0.0014 | 85.9303±0.0731 | 117.9925±0.0172 |
| MPPDE | 8.9810±0.0014 | 12.1595±0.0022 | 20.7453±0.0008 | 32.1098±0.0018 | 54.2387±0.0006 | 90.0190±0.0007 |
| NFS | **2.1992**±0.0021 | **2.8280**±0.0033 | **3.9178**±0.0054 | **5.0182**±0.0080 | **4.7865**±0.0042 | **6.1384**±0.0069 |

Table B7: Performance comparison with graph spatio-temporal benchmarks ($n_t = 10$).

| Graph Spatio-Temporal Models | MAE ($\times 10^{-3}$) | RMSE($\times 10^{-3}$) | MAE($\times 10^{-3}$) | RMSE($\times 10^{-3}$) | MAE($\times 10^{-3}$) | RMSE($\times 10^{-3}$) |
|---|---|---|---|---|---|---|
| | Burgers' ($n_s = 512, n'_t = 10$) | | Burgers' ($n_s = 256, n'_t = 20$) | | Burgers' ($n_s = 512, n'_t = 40$) | |
| DCRNN | 2.6122±0.0014 | 3.8435±0.0019 | 4.6126±0.0015 | 6.8853±0.0033 | 8.5880±0.0020 | 12.7394±0.0037 |
| AGCRN | 4.6667±0.0001 | 6.2791±0.0001 | 10.4900±0.0009 | 13.9810±0.0022 | 15.6143±0.0002 | 21.0937±0.0001 |
| GCGRU | 1.6643±0.0002 | 2.5074±0.0003 | 3.1400±0.0010 | 4.8008±0.0019 | 5.7653±0.0021 | 8.9335±0.0028 |
| MPPDE | 1.1271±0.0004 | 1.8838±0.0007 | 2.4554±0.0003 | 4.4315±0.0006 | 4.1213±0.0006 | 6.1980±0.0009 |
| NFS | **0.1085**±0.0016 | **0.1504**±0.0021 | **0.1634**±0.0018 | **0.2328**±0.0018 | **0.1983**±0.0001 | **0.2775**±0.0003 |
| | KdV ($n_s = 512, n'_t = 10$) | | KdV ($n_s = 256, n'_t = 20$) | | KdV ($r = 512, n'_t = 40$) | |
| DCRNN | 2.3196±0.0001 | 4.1634±0.0001 | 3.4503±0.0005 | 5.7450±0.0003 | 4.9286±0.0010 | 8.3912±0.0008 |
| AGCRN | 3.9350±0.0001 | 6.1166±0.0001 | 5.6631±0.0001 | 8.1191±0.0001 | 8.2893±0.0002 | 11.5684±0.0003 |
| GCGRU | 1.6643±0.0001 | 2.5074±0.0001 | 3.4205±0.0001 | 5.6873±0.0001 | 2.5032±0.0002 | 5.4515±0.0003 |
| MPPDE | 1.4967±0.0003 | 2.6309±0.0002 | 2.9708±0.0027 | 5.3811±0.0050 | 2.4293±0.0006 | 4.9310±0.0005 |
| NFS | **0.0816**±0.0012 | **0.1512**±0.0022 | **0.1576**±0.0007 | **0.3114**±0.0018 | **0.3210**±0.0021 | **0.6873**±0.0049 |
| | NS ($n_s = 4096, n'_t = 10$) | | NS ($n_s = 1024, n'_t = 20$) | | NS ($n_s = 4096, n'_t = 40$) | |
| DCRNN | 8.7025±0.0003 | 12.5238±0.0002 | 27.1069±0.0024 | 39.1259±0.0031 | 59.6602±0.0177 | 88.2946±0.0146 |
| AGCRN | OOM | OOM | 42.4197±0.0006 | 60.5375±0.0008 | OOM | OOM |
| GCGRU | 6.3570±0.0001 | 9.7306±0.0002 | 21.3537±0.0026 | 32.9674±0.0033 | 57.2493±0.0085 | 84.1847±0.0106 |
| MPPDE | 5.4353±0.0041 | 7.8838±0.0037 | 17.5902±0.0013 | 25.9372±0.0016 | 42.3057±0.0066 | 76.3374±0.0069 |
| NFS | **0.9335**±0.0011 | **1.3254**±0.0012 | **1.8239**±0.0012 | **2.5291**±0.0008 | **3.2768**±0.0026 | **4.3988**±0.0009 |

### B.4 MORE VISUALIZATION

Here we provide more visualization results on the three equations. See Fig. B2, Fig. B3 and Fig. B4.

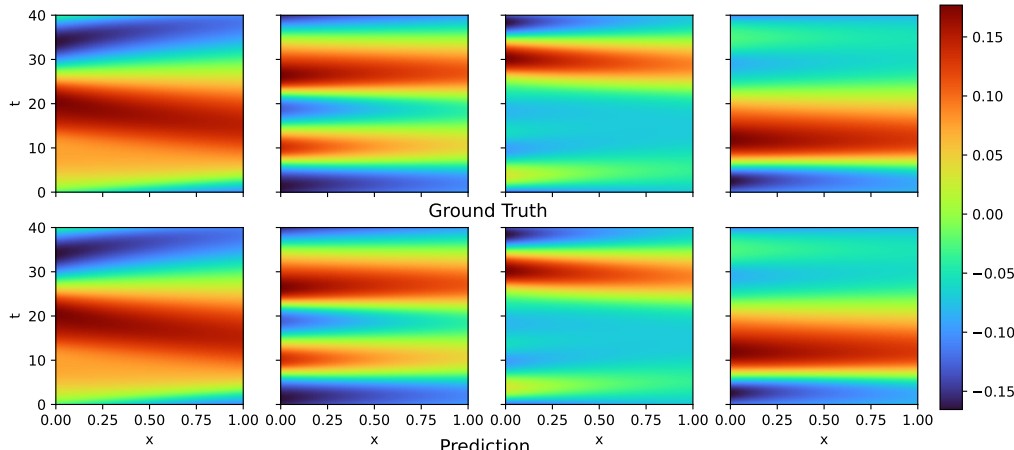

Figure B2: Visualization on equispaced Burgers' equation.

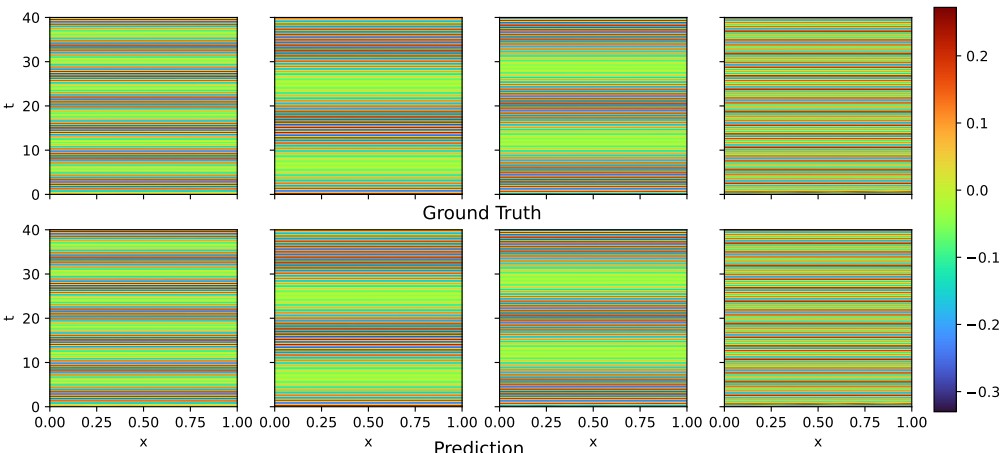

Figure B3: Visualization on equispaced KdV equation.

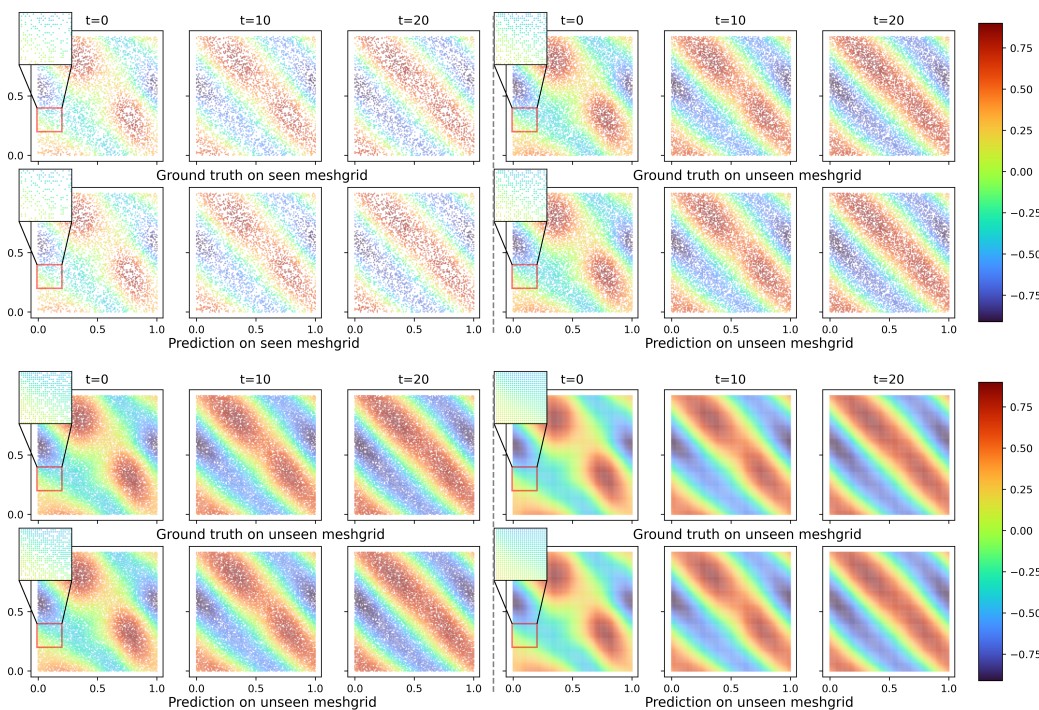

Figure B4: Visualization on non-equispaced NS equation: The training mesh ($n_s = 4096$ in upper-left) is different from the meshes in inference process ($n'_s = 8192$ in upper-right, $n'_s = 12288$ in lower-left and $n's = 16384$ in lower-right).

## B.5 MESH-INVARIANT EVALUATION

Table B8: Mesh-invariant performance of NFS on Burgers' and KdV equations ($n_t = 10$).

| | MAE ($\times 10^{-3}$) | RMSE($\times 10^{-3}$) | MAE($\times 10^{-3}$) | RMSE($\times 10^{-3}$) |
|---|---|---|---|---|
| | Burgers' ($n_s = 512, n_t = 10, n'_t = 40$) | | KdV ($n_s = 512, n_t = 10, n'_t = 40$) | |
| $\boldsymbol{X}$ | $0.1983_{\pm 0.0001}$ | $0.2775_{\pm 0.0002}$ | $0.3210_{\pm 0.0021}$ | $0.6873_{\pm 0.0049}$ |
| $n'_s = 1.3 n_s$ | $0.2371_{\pm 0.0034}$ | $0.3143_{\pm 0.0041}$ | $0.3769_{\pm 0.0030}$ | $0.7805_{\pm 0.0077}$ |
| $n'_s = 1.7 n_s$ | $0.2898_{\pm 0.0113}$ | $0.3742_{\pm 0.0102}$ | $0.4084_{\pm 0.0072}$ | $0.8419_{\pm 0.0174}$ |
| $n'_s = 2.0 n_s$ | $0.3052_{\pm 0.0098}$ | $0.4180_{\pm 0.0100}$ | $0.4111_{\pm 0.0042}$ | $0.8471_{\pm 0.0074}$ |

Table B9: Performance of NFS with its variants of NS equations ($n_t = 10$) on unseen meshes.

| Flex + LN | MAE ($\times 10^{-3}$) | RMSE($\times 10^{-3}$) | MAE($\times 10^{-3}$) | RMSE($\times 10^{-3}$) | MAE($\times 10^{-3}$) | RMSE($\times 10^{-3}$) |
|---|---|---|---|---|---|---|
| | NS ($n_s = 4096, n'_t = 10$) | | NS ($n_s = 1024, n'_t = 20$) | | NS ($n_s = 4096, n'_t = 40$) | |
| $\boldsymbol{X}$ | $0.9335_{\pm 0.0011}$ | $1.3254_{\pm 0.0012}$ | $1.8239_{\pm 0.0012}$ | $2.5291_{\pm 0.0008}$ | $3.2768_{\pm 0.0026}$ | $4.3988_{\pm 0.0009}$ |
| $n'_s = 2 n_s$ | $0.9731_{\pm 0.0034}$ | $1.5042_{\pm 0.0057}$ | $2.3530_{\pm 0.0051}$ | $3.3320_{\pm 0.0074}$ | $3.5439_{\pm 0.0085}$ | $4.7904_{\pm 0.0168}$ |
| $n'_s = 3 n_s$ | $1.1071_{\pm 0.0021}$ | $1.5716_{\pm 0.0038}$ | $2.5179_{\pm 0.0089}$ | $3.5477_{\pm 0.0125}$ | $3.6584_{\pm 0.0180}$ | $4.8858_{\pm 0.0246}$ |
| $n'_s = 4 n_s$ | $1.1015_{\pm 0.0000}$ | $1.5627_{\pm 0.0000}$ | $2.5919_{\pm 0.0064}$ | $3.6526_{\pm 0.0071}$ | $3.6608_{\pm 0.0000}$ | $4.9521_{\pm 0.0000}$ |

| Gaus + LN | MAE ($\times 10^{-3}$) | RMSE($\times 10^{-3}$) | MAE($\times 10^{-3}$) | RMSE($\times 10^{-3}$) | MAE($\times 10^{-3}$) | RMSE($\times 10^{-3}$) |
|---|---|---|---|---|---|---|
| | NS ($n_s = 4096, n'_t = 10$) | | NS ($n_s = 1024, n'_t = 20$) | | NS ($n_s = 4096, n'_t = 40$) | |
| $\boldsymbol{X}$ | $1.6341_{\pm 0.0034}$ | $2.1992_{\pm 0.0042}$ | $2.1976_{\pm 0.0065}$ | $3.0219_{\pm 0.0090}$ | $3.6422_{\pm 0.0026}$ | $5.0097_{\pm 0.0039}$ |
| $n'_s = 2 n_s$ | $2.8589_{\pm 0.0062}$ | $4.0562_{\pm 0.0126}$ | $3.7465_{\pm 0.0041}$ | $5.1308_{\pm 0.0097}$ | $3.9092_{\pm 0.0041}$ | $5.2402_{\pm 0.0075}$ |
| $n'_s = 3 n_s$ | $3.4513_{\pm 0.0168}$ | $4.5199_{\pm 0.0377}$ | $5.7712_{\pm 0.0123}$ | $5.7137_{\pm 0.0199}$ | $4.2102_{\pm 0.0082}$ | $5.5057_{\pm 0.0138}$ |
| $n'_s = 4 n_s$ | $3.4357_{\pm 0.0000}$ | $4.7382_{\pm 0.0000}$ | $5.5990_{\pm 0.0066}$ | $5.5958_{\pm 0.0049}$ | $4.2628_{\pm 0.0000}$ | $5.7679_{\pm 0.0000}$ |

| Flex + ~~LN~~ | MAE ($\times 10^{-3}$) | RMSE($\times 10^{-3}$) | MAE($\times 10^{-3}$) | RMSE($\times 10^{-3}$) | MAE($\times 10^{-3}$) | RMSE($\times 10^{-3}$) |
|---|---|---|---|---|---|---|
| | NS ($n_s = 4096, n'_t = 10$) | | NS ($n_s = 1024, n'_t = 20$) | | NS ($n_s = 4096, n'_t = 40$) | |
| $\boldsymbol{X}$ | $1.2138_{\pm 0.0030}$ | $1.7293_{\pm 0.0047}$ | $2.5119_{\pm 0.0036}$ | $3.4923_{\pm 0.0058}$ | $4.2083_{\pm 0.0037}$ | $5.6761_{\pm 0.0092}$ |
| $n'_s = 2 n_s$ | $1.4882_{\pm 0.0146}$ | $2.1681_{\pm 0.0300}$ | $7.0203_{\pm 0.0203}$ | $10.6096_{\pm 0.0345}$ | $5.8975_{\pm 0.0060}$ | $8.7704_{\pm 0.0189}$ |
| $n'_s = 3 n_s$ | $1.6384_{\pm 0.0088}$ | $2.4130_{\pm 0.0169}$ | $7.9177_{\pm 0.0059}$ | $11.9825_{\pm 0.0118}$ | $6.6622_{\pm 0.0063}$ | $9.5874_{\pm 0.0131}$ |
| $n'_s = 4 n_s$ | $1.6975_{\pm 0.0000}$ | $2.5008_{\pm 0.0000}$ | $7.1962_{\pm 0.0101}$ | $10.8860_{\pm 0.0098}$ | $6.6951_{\pm 0.0000}$ | $9.6334_{\pm 0.0000}$ |

The mesh-invariant evaluation on Burgers' and KDV Equations of NFS are given in Table. B8. In Table. B8, when the spatial resolution is just 512, inference performance on unseen meshes deteriorates. This result also validate our conclusion *(2)* in the third paragraph in Sec. 4.

Besides, we give a full evaluation on mesh-invairance of NFS in NS equation, with its variants as a detailed results corresponding to Table. B9.

### B.6 NEIGHBORHOOD SIZE'S EFFECTS

The effects of mean neighborhood size on the predictive performance on `Burgers'` $(n_s = 512, n_t = 10, n'_t = 40)$ and `KDV` $(n_s = 512, n_t = 10, n'_t = 40)$ are shown in Fig. B5.

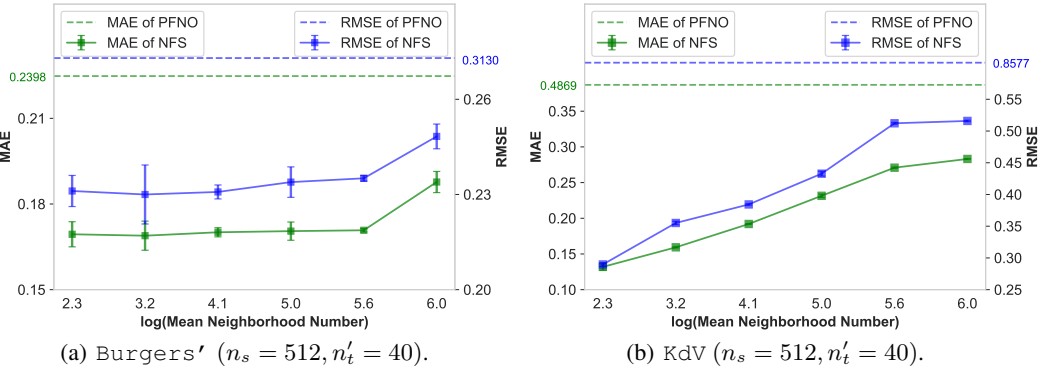

(a) `Burgers'` $(n_s = 512, n'_t = 40)$.      (b) `KdV` $(n_s = 512, n'_t = 40)$.

Figure B5: The change of MAE and RMSE of NFS with the increase of neighborhood size on `Burgers'` $(n_s = 512, n_t = 10, n'_t = 40)$ and `KdV` $(n_s = 512, n_t = 10, n'_t = 40)$. PFNO is the baseline.

### B.7 INTERPOLATION WITH OTHER VISION MIXERS

We conduct experiments on non-equispaced NS equations with the combination of our interpolation layers and other Vision Mixers to figure out if they can achieve camparable performance.

Table B10: Performance of different Vision Mixers combined with the interpolation layers in non-equispaced scenarios on NS equations $(n_t = 10)$.

| | MAE $(\times 10^{-3})$ | RMSE $(\times 10^{-3})$ | MAE $(\times 10^{-3})$ | RMSE $(\times 10^{-3})$ | MAE $(\times 10^{-3})$ | RMSE $(\times 10^{-3})$ |
|---|---|---|---|---|---|---|
| VIT | NS $(n_s = 4096, n'_t = 10)$ | | NS $(n_s = 1024, n'_t = 20)$ | | NS $(n_s = 4096, n'_t = 40)$ | |
| $X$ | OOM | OOM | OOM | OOM | OOM | OOM |
| | MAE $(\times 10^{-3})$ | RMSE $(\times 10^{-3})$ | MAE $(\times 10^{-3})$ | RMSE $(\times 10^{-3})$ | MAE $(\times 10^{-3})$ | RMSE $(\times 10^{-3})$ |
| MLPMIXER | NS $(n_s = 4096, n'_t = 10)$ | | NS $(n_s = 1024, n'_t = 20)$ | | NS $(n_s = 4096, n'_t = 40)$ | |
| $X$ | $6.1854_{\pm 0.0012}$ | $8.1556_{\pm 0.0018}$ | $9.4593_{\pm 0.0028}$ | $12.1316_{\pm 0.0022}$ | $10.1862_{\pm 0.0045}$ | $13.1548_{\pm 0.0051}$ |
| $n'_s = 2n_s$ | $8.1573_{\pm 0.0126}$ | $11.2258_{\pm 0.0147}$ | $12.0706_{\pm 0.0132}$ | $14.9460_{\pm 0.0159}$ | $10.6003_{\pm 0.0127}$ | $13.6872_{\pm 0.0238}$ |
| $n'_s = 3n_s$ | $8.1952_{\pm 0.0088}$ | $11.3840_{\pm 0.0171}$ | $14.9910_{\pm 0.0094}$ | $17.8415_{\pm 0.0110}$ | $10.5633_{\pm 0.0140}$ | $13.6394_{\pm 0.0147}$ |
| $n'_s = 4n_s$ | $8.7773_{\pm 0.0000}$ | $11.3313_{\pm 0.0000}$ | $14.9517_{\pm 0.0125}$ | $17.7857_{\pm 0.0199}$ | $10.5414_{\pm 0.0000}$ | $13.6106_{\pm 0.0000}$ |
| | MAE $(\times 10^{-3})$ | RMSE $(\times 10^{-3})$ | MAE $(\times 10^{-3})$ | RMSE $(\times 10^{-3})$ | MAE $(\times 10^{-3})$ | RMSE $(\times 10^{-3})$ |
| GFN | NS $(n_s = 4096, n'_t = 10)$ | | NS $(n_s = 1024, n'_t = 20)$ | | NS $(n_s = 4096, n'_t = 40)$ | |
| $X$ | $12.2373_{\pm 0.0091}$ | $16.2902_{\pm 0.0133}$ | $10.2768_{\pm 0.0084}$ | $13.7852_{\pm 0.0078}$ | $14.7765_{\pm 0.0055}$ | $19.4872_{\pm 0.0106}$ |
| $n'_s = 2n_s$ | $13.7752_{\pm 0.0164}$ | $18.3108_{\pm 0.0181}$ | $17.7216_{\pm 0.0225}$ | $24.1397_{\pm 0.0371}$ | $15.9083_{\pm 0.0235}$ | $21.0041_{\pm 0.0256}$ |
| $n'_s = 3n_s$ | $13.7054_{\pm 0.0122}$ | $18.2192_{\pm 0.0184}$ | $17.8238_{\pm 0.0112}$ | $24.2783_{\pm 0.0196}$ | $15.8986_{\pm 0.0156}$ | $20.9943_{\pm 0.0158}$ |
| $n'_s = 4n_s$ | $13.7140_{\pm 0.0000}$ | $18.2271_{\pm 0.0000}$ | $17.8207_{\pm 0.0105}$ | $24.2833_{\pm 0.0141}$ | $15.8736_{\pm 0.0000}$ | $20.9772_{\pm 0.0000}$ |

## B.8 COMPLEXITY COMPARISON

We here first give Table. B11 to show the complexity of time and memory of all the evaluated methods on NS ($r = 64, n_t = 10, n'_t = 40$).

Table B11: comparison on complexity of the evaluated methods

| Type | Methods | Time/Epoch | Peak Memory | Parameter Number |
|---|---|---|---|---|
| Graph Spatio-Temporal Model | GCGRU | $6'18''$ | 8660MB | 74945 |
| | DCRNN | $9'38''$ | 11120MB | 148673 |
| | AGCRN | OOM | OOM | OOM |
| | MPPDE | $10'54''$ | 23333MB | 622161 |
| Vision Mixer | VIT | $3'14''$ | 32166MB | 773217 |
| | MLPMIXER | $1'12''$ | 4421MB | 79749953 |
| | GFN | $48''$ | 3296MB | 1361729 |
| | FNO | $27''$ | 3748MB | 6299425 |
| | PFNO | $43''$ | 3380MB | 9742145 |
| | NFS | $2'02''$ | 31938MB | 37891937 |

Table B12: Detailed complexity of NFS

| Interpolation on Resampled Points | | |
|---|---|---|
| Neighbor Searching | Kernel Calculation | Weighted Summation |
| 3522MB | 3102MB | 6884MB |
| Interpolation back on Original Points | | |
| Neighbor Searching | Kernel Calculation | Weighted Summation |
| 2506MB | 2754MB | 6884MB |

It demonstrates that our method has comparable efficiency to Vision Mixers. For the graph spatial-temporal models, they suffer from the recurrent network structures and thus are extremely time-consuming while the parameter number is small, limiting their flexibility.

**Time.** However, once we compare the used time in PFNO and NFS, we will find that the interpolation layers are considerably time-consuming. Another module that cost time complexity is the normalization layer, as the original FNO does not include Layer-Norm in its architecture, but it is stacked in PFNO. Theoretically, PFNO handles down-sampled grids in a low resolution, because of the patchwise embedding. However, it takes more time than FNO. Therefore, we conclude that the time complexity brought from Layer-Norm is very significant, but it is affordable because of the performance improvements.

**Memory.** Besides, the operation of searching for each spatial point's neighborhood and calculating weighted summation in Eq. (9) and Eq. (10) are very memory-consuming. We test it on the same experiment, and give the memory usage of different models in forward process, as shown in Table. B12. The memory cost in backward process is 6902MB.

## C EMPIRICAL OBSERVATION FOR THEOREM 3.1

In Sec. 3.2, Theorem 3.1 is proved to assure the expressivity of NFS. However, no further evidence gives the assurance of the convergence of the kernel interpolation. Here we conduct empirical study to give some clues.

We conduct experiments on NS equation with $n_s = 4096, n_t = 10, n'_t = 40$. In a single trial, NFS is trained with fixed meshes. We repeated the trials 10 times with different meshes, and then give the

one-v.s.-all deviations of the representation states calculated by

$$\text{Diff} = \frac{1}{90} \sum_{i \neq j, i, j = 1}^{10} \frac{1}{m_s, n'_t} || \frac{|H_i - H_j|}{|H_i| + |H_j|} ||_1,$$

where $H_i$ is the representation states of the shape $[\sqrt{m_s}, \sqrt{m_s}, n'_t]$, and $|\cdot|$ is the element-wise absolute value, and $|| \cdot ||_1$ is the 1-norm of the matrix. If the Diff is small in the beginning and end, it can be inferred that the interpolation kernel function converges to a similar mapping since the final predictions are close to ground truth in these experiments, and the inputs are sampled from the same instance of PDEs. We give the Diff before the first FNO and after the final of FNO layers in Table. C1. The small values indicate that the trained model usually has similar representation states. Figure. C1 and C2 give visualizations of representation states obtained by one instance of NS equation in two different trials. It indicates that the differences are getting smaller during the training.

Table C1: The defined Diff calculated by different epochs.

| Epoch | $\text{Diff}_{\text{begin}}$ | $\text{Diff}_{\text{end}}$ |
|-------|------------|----------|
| 0     | 0.0676     | 0.0978   |
| 500   | 0.0102     | 0.0353   |

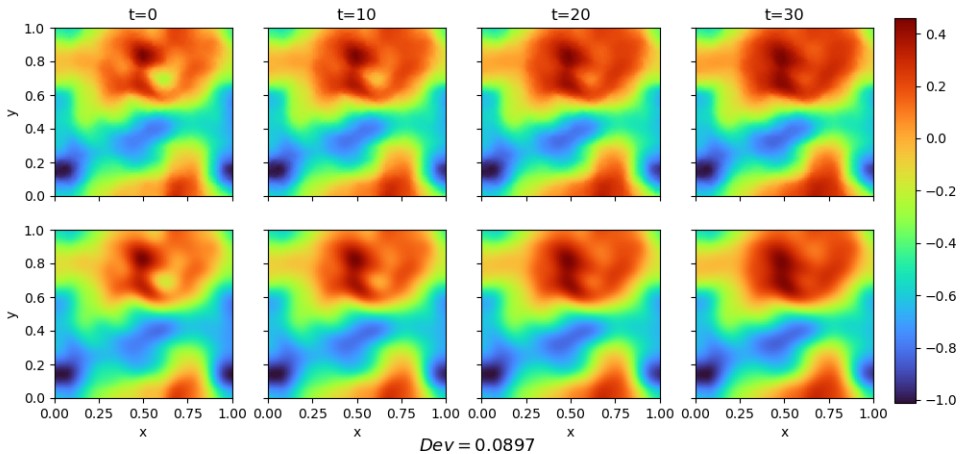

(a) Representation states at the beginning of FNO layers in two trials of Epoch 0

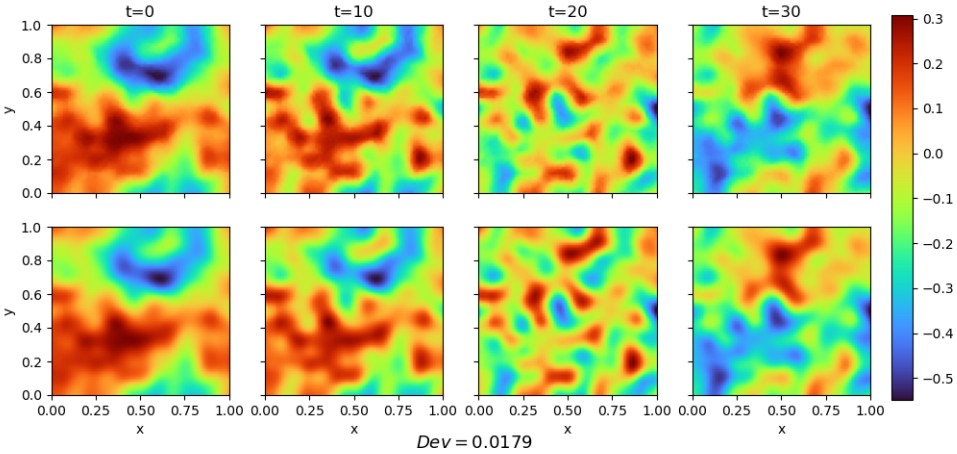

(b) Representation states at the beginning of FNO layers in two trials of Epoch 500

Figure C1: Visualization on different representation states at the beginning of FNO layers.

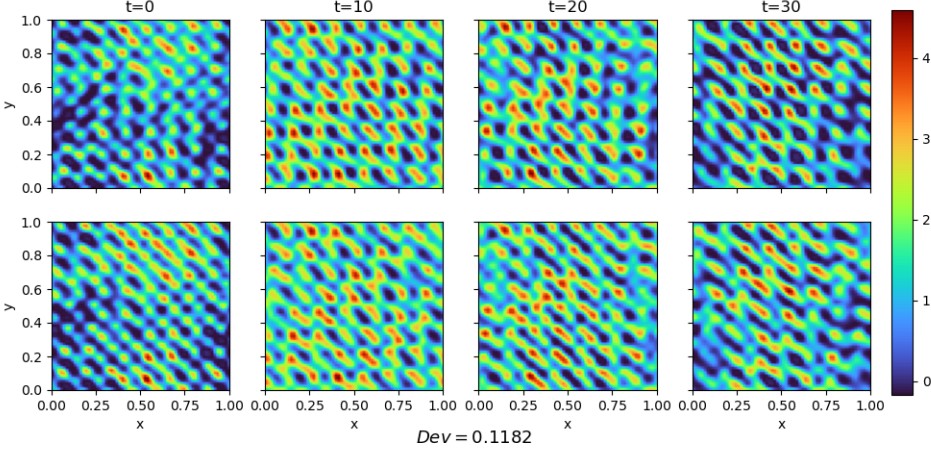

(a) Representation states in the end of FNO layers in two trials of Epoch 0

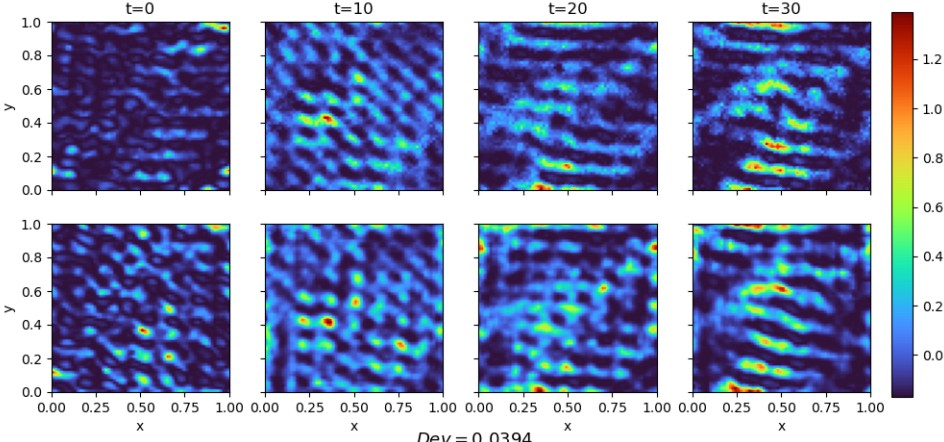

(b) Representation states in the end of FNO layers in two trials of Epoch 500

Figure C2: Visualization on different representation states in the end of FNO layers.

As a result, we present the one-v.s.-all differences of different epochs in the training process, to validate the convergence, as shown in Figure. C3.

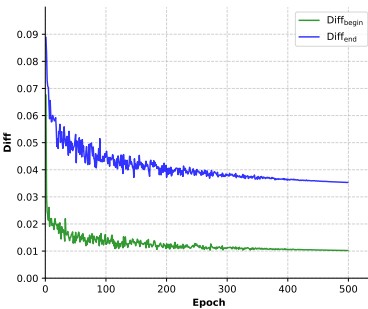

Figure C3: Convergence of Diff.

