# OpenReview forum: "Non-equispaced Fourier Neural Solvers for PDEs"
_ICLR.cc/2023/Conference — Submitted to ICLR 2023_

### Official Review · Reviewer_gMaM · 2022-10-27

**Confidence:** 4
**Correctness:** 2
**Technical Novelty And Significance:** 2
**Empirical Novelty And Significance:** 2
**Recommendation:** 3

**Clarity, Quality, Novelty And Reproducibility:**

The presentation is quite unclear. There is also some issue with the novelty here since DeepONets are anyway a superset of FNO and they work fine with randomly sampled data. So where is the evidence that this new form of FNO will beat what DeepONets anyway do?

Also the testing is extremely weak since consistently the authors seem to be working in the regime of test data being much smaller than training data. This is very problematic - because no evidence is shown that the risk estimates have even converged for such small test data sizes.

**Strength And Weaknesses:**

There is a total lack of transparency in this paper as to what is the new loss function - and this ambiguity starts from not having any clarity about how is a supposed interpolation being done from the sampled non-equispaced points to a new sample of equispaced points.  I believe that the $\tilde{\cal F}$ in equation 7 and 8 is what is replacing the ${\cal F}$ in equation 4. But this is only my guess and this still doesn't really define what is the new replacement of the basic equation 3 of FNO.

Equations 6,7,8 and the periodic heat kernel are all seemingly defined in terms of their resampled equispaced points. Its entirely as to where and how the given non-equispaced points got used.


**Summary Of The Paper:**

This paper raises an important question as to how one can do operator learning when equispaced discretizations of the functions are not available. But the supposed solution proposed in this paper is quite unconvincing.

**Summary Of The Review:**

I believe there are very serious issues with the paper and hence my rating is on the lower side.

---

> ### Author Response · Authors · 2022-11-06
> **Response**
>
> From your comments, we question whether you have a foundation and background in neural operators or even machine learning. Please  be responsible to read the full paper and show your professionality.
>
> 1.First, on your confusion about equation 6.7.8:
> We here explain the Equations in detail for your better understanding. Note that the description is mainly based on  some basic knowledge of functional analysis, signal process, and neural operators that an undergraduate student should be familiar with.
> * Equation 6:
> $f: D \rightarrow \mathbb{R}^{d}$, so we denote it as $f \in \mathcal{U}(D; \mathbb{R}^{d})$ where $\mathcal{U}(D; \mathbb{R}^{d})$ is a Banach space of function.
> $\mathcal{H}: \mathcal{U}(D; \mathbb{R}^{d}) \rightarrow \mathcal{U}(D; \mathbb{R}^{d})$ is an interpolation operator. If we write $f_{\mathrm{org}}(x) = \sum_{i=1}^{n_s} f(x)\delta_{x_{i}}(x)$ in a sampling forms, where $\delta_{y}(x) = 1$ if $x=y$ else $0$, and {$x_i :1\leq i \leq n_s$} is the original spatial points. Then, $f_{\mathrm{int}}(x) = \mathcal{H}(f_{\mathrm{org}})(x) = \sum_{j=1}^{m_s}\sum_{i=1}^{n_s}f(x_i)h_{\tau}(x_i-x)\delta_{x_j}(x) \approx \sum_{j=1}^{m_s}f(x)\delta_{x_j}(x)$, and {$x_j :1\leq j \leq m_s$ }is the resampled spatial points. Therefore, $ \mathcal{H}(f)(x_j) = \sum_{i=1}^{n_s}f(x_i)h_{\tau}(x_i-x_j)$. These are the details. All these formulations can be referred to Sec. 3.1 and Sec. 2.1, except that we use a formulation of sampling functions to help you to understand.
>
> * Equation 7 and 8:
> First, replace $f$ from Equation 6 with $v$. Then, a difference is that in Equation 6, $\mathcal{H}$ is a linear operator, as $\mathcal{H}: \mathcal{U}(D; \mathbb{R}^{d}) \rightarrow \mathcal{U}(D; \mathbb{R}^{d})$, so $\mathcal{H} \in \mathcal{L}(\mathcal{U})$ in Equation 6, but in Equation 7 and 8, $\mathcal{H}$ is an interpolation operator mapping, as $\mathcal{H}: \mathcal{A} \rightarrow \mathcal{L}(\mathcal{U})$. Therefore, $\mathcal{H}(a) \in \mathcal{L}(\mathcal{U})$, and $\mathcal{H}(a)v$ is the function after the interpolation from non-equispaced {$x_i$} to equispaced {$x_j$}. These details are also given in Sec. 3.2, and we hope you can understand them mathematically.
>
>     By this mean, $\mathcal{\hat F}$ can be decomposed into two parts, one is interpolation operator $\mathcal{H}$ or $\mathcal{H}(a)$, and the second is equispaced FFT $\mathcal{F}$. As a result, it is different from the basic equation of FNO, as its variant to non-equispaced scenarios. If you still cannot understand, we recommend you ask me or other reviewers in the discussion procedures.
>
> 2. Second, we disagree with your irresponsible claim that DeepONets are anyway a superset of FNO:
> *(i) Vanilla DeepONet is formulated in general as
> $G(v)(x) = \sum_{k} Br_k(v) Tr_k(x)$, where $\{Br_k\}$ are the Branch Nets and $\{Tr_k\}$ are the Trunk Net. They are implemented by fully connected neural networks. However, FNO reads $G’(v)(x) = \mathcal{F}^{-1}(R\cdot\mathcal{F}(v))(x)$, where $R$ are learnable parameters. The implementation is different: one uses FFT and convolution theorems to map function $v$ to its solution, and the other use MLPs.
> *(ii)If they can be merged in DeepONet framework, why the author of DeepONet proposed another paper to compare the two architectures of DeepONet and FNO? (https://arxiv.org/abs/2111.05512)
> We guess you would like to claim that DeepONet and FNO are all Neural Operators, as proposed in Equation 30 and Proposition 5 in Sec. 5.1 in https://arxiv.org/abs/2108.08481. If so, please comment rigorously.
>
> 3. Third, where is the evidence that this new form of FNO will beat what DeepONets anyway do?
> * Please refer to General Response on Comparison with DeepONet.
> 4. To respond to your comments on our ‘very problematic’ test procedures: `Also the testing is extremely weak ... test data being much smaller than training data.’
>
> * Here we list the training set sizes and test set sizes as the experimental protocols used in benchmark Neural Operators and ours in Table 2 in General Response. As shown, only the vallina DeepONet uses more test samples than training, because the evaluation of it is based on a dataset consisting of 40 spatial points, not such high-resolution ones in recent years. We claim that smaller sizes of test samples than training ones are usually recognized as Machine-Learning experimental protocols, and the protocol we use is not problematic and `extremely’ weak. Besides, this makes us wonder if you have ever read other neural operator works, or even run their codes. If you only accept experimental protocol in DeepONet as the gold standard, we hope you open your mind and dive into more recent research.
>
> Besides, we suppose that you are curious about what if the test set size is larger. We here generated more samples and conducted experiments to give the metrics in Table.3 in General Response on different test sizes in the NS equations, to eliminate your doubts.

---

> > ### Comment · Area_Chair_JvZs · 2022-11-06
> > **Please reconsider**
> >
> > Hi authors,
> >
> > a quick note (orthogonal from the scientific merits of your response): please refrain from ad hominem attacks on reviewers. This is not keeping in line with the ICLR code of conduct (https://iclr.cc/public/CodeOfConduct). If you have further questions, feel free to please message me or the program chairs.
> >
> >  -AC

---

> > > ### Author Response · Authors · 2022-11-06
> > > **response to AC**
> > >
> > > We are sorry that the words we use are kind of aggressive.
> > > However, if what we state is truth, please ignore some negative comments without any supports. And respect the value of our works rather than take baseless criticism as grounds for rejection.
> > >
> > > The response will be reorganized thanks to your remind.
> > >
> > > For other complaints about the review quality, we will directly report to you. Thanks

---

### Official Review · Reviewer_Y9dA · 2022-10-28

**Confidence:** 4
**Correctness:** 4
**Technical Novelty And Significance:** 2
**Empirical Novelty And Significance:** 3
**Recommendation:** 5

**Clarity, Quality, Novelty And Reproducibility:**

Clarity: This work is clear. Some notations such as d_a and d_u and for that matter d could have been explained a bit more... but I would understand if the authors skipped elaborating a bit more due to page limit constraints. At the same time, the paper seems to be written in a bit of a rush (naturally expected, but have to make a note of it here).

Quality: The work is novel. But, is it novel enough to publish in ICLR, just on the merit of significant results compared to FNO? I don't think so. Nevertheless, the authors did a commendable job presenting the results and all comparisons with other methods, etc. But that still doesn't give an impact in terms of a paper in ICLR.

Originality: this work is original.

**Strength And Weaknesses:**

+ better memory efficiency over FNOs

- It still has all several issues plagued with FNO as well.
- The comparisons are done with standard vision mixers and other models. But methods such as DeepONets are cited but not compared against.
- the theory (theorem 3.1) provided is not giving any insights but is only provided to prove something that the authors don't relate with the results obtained. It's as if the theory is not meant to prove anything but that approximations exist. I did a random search of where theorem 3.1 is cited in the whole paper and didn't find any references in the results or the rest of the main paper.

**Summary Of The Paper:**

In this paper, the authors propose a new approach for solving PDEs using non-equispaced Fourier Neural Operators. The authors draw some analogs between vision mixer models and FNO and then build over it to obtain the non-equispaced FNO. For the non-equispaced FNO, the authors use a non-equispaced interpolation function that helps construct an equispaced signal and thus helps use FNO for the PDE solving. The idea is novel, and the results show improvement over FNOs, and further, they also overcome memory issues to some extent.

**Summary Of The Review:**

The work proposed by the authors is certainly novel; the paper proposed a non-equispaced Fourier Neural PDE solver, and the contributions include (I) some connection between vision mixers and FNOs (ii) the development of NEI layer to obtain non-equispaced FNO (iii) significant results compared to baseline.

The contributions are somewhat like incremental research with minimal improvement over the SoTA. Hence, I recommend a weak reject. What could convince me is (i) some more comparison with standard neural PDE solvers (ii) Some more theory on what kind of sampling strategies one should try or something (iii) and/or some more results which can be related to the theory proposed.

---

> ### Author Response · Authors · 2022-11-06
> **Response to Reviewer Y9dA:**
>
> Thank for your recognition of our works, and your comments with deep insights. Here are some responses to your doubts.
> 1. On more benchmark comparison:
> * We gave **experimental results on DeepONet in General Response**, and will update these results in our newly-version paper. If you have any other concerns about the methods we have not compared with, please point them out, and as long as their code is open, and could handle PDEs in our papers, we will try to conduct experiments A.S.A.P.
> We hope the experimental comparison can convince you of your concern (i).
>
> 2. On corresponding results to the theorem:
> * As you can see, the context of our paper is very sufficient, so that some representation state analysis is not appended. Some experiments have been conducted on the convergence of representation states to figure out whether the interpolation kernel converges to a mapping. **The details are given in the newly updated Appendix C.**
>
> * To be specific, we trained our NFS with fixed meshes in one trial, and repeated the trials 10 times with different meshes, and then we give the one-v.s.-all differences of the representation states calculated by
> $$Diff = \frac{1}{90}\sum_{i\neq j, i,j=1}^{10}\frac{1}{m_s n_t}|| \frac{|H_{i} - H_{j}|}{|H_{i}| + |H_{j}|}||,$$
> where $H_{i}$ is the representation states of the shape $[\sqrt{m_s}, \sqrt{m_s}, n'_t]$, and $|\cdot|$ is the element-wise absolute value, and $||\cdot||$ is the 1-norm of the matrix.
> **If the $ Diff$ is small before the first FNO and after the final of FNO layers, it can be inferred that the interpolation kernel function converges to a similar mapping** since the final predictions are close to ground truth in these experiments, and the inputs are sampled from the same instance of PDEs.
> Therefore, we first give some visualization of 2D representation states of one single instance in different trials and the calculated MAE in Appendix C. For the page limit, the corresponding analysis to Universal Approximation Theorem in the context will be added in the next version.
>
> * Second, we draw the Diff on the test set according to the first and final representation states to show the convergence (Appendix C). This is a very interesting observation, and we hope these observations can convince you of your concern (iii). It is also a demonstration of our NFS’s ability to generalize to unseen meshes.
>
> 3. On the proposition of more theories.
> As you acknowledge, our paper focuses more on the **empirical study over theoretical contributions**, and theoretical research on Neural Operators is another line of direction.
>
>     (i) First, **the theorem** proposed is to assure the universal approximation ability of the NEI layers, which assures NFS’s expressivity as a key that **is often overlooked in similar works**. For example, in most works such as MGKN and FNO, the assurance of expressivity is not proved.
>
>     (ii) Besides, **the approximation theorem is what most current theoretical work focuses on**. For example, in the studies as theoretical analysis on Neural Operators, such as  http://tensorlab.cms.caltech.edu/users/anima/pubs/GraphPDE_Journal.pdf , it uses several pages (Sec. 4) to prove the universal approximation theorem of the proposed architectures of Neural Operators.
>
>     We are sorry that more theoretical contributions may not be made in this short paper. However, it is a good direction of research interest, such as sampling strategy and converging rate. We would like to call the theorem an accessory bonus. And we hope the statements can convince you of your concern (ii).
>
> Other comments:
>
>    ` But, is it novel enough to publish in ICLR, just on the merit of significant results compared to FNO?’
>
> * As you can see from the benchmark comparison, FNO may not generate significantly remarkable results compared with other Vision Mixers. And our work combines them to allow (i) remarkable experimental accuracy; (ii) extension to non-equispaced scenarios.
>
> ` The paper seems to be written in a bit of a rush.’
>
> * Our paper is written in a style closer to Computer-Vision, please refer to our response on `Novelty’ to Reviwer Gvhj to get more clues on writing. We have to declare many aspects including mesh-invariance, Neural Operator, Vision Mixer, and Graph-based Spatio-Temporal Models in a single paper, so many aspects cannot be expressed in detail. To address it, lots of references are included in our paper, we hope the writing style can help the readers follow our logic and motivation easily.

---

### Official Review · Reviewer_Gvhj · 2022-10-28

**Confidence:** 5
**Correctness:** 4
**Technical Novelty And Significance:** 2
**Empirical Novelty And Significance:** 3
**Recommendation:** 6

**Clarity, Quality, Novelty And Reproducibility:**

The paper is well-written and supported by complete experiments. However, the method itself is not ver novel. The kernel interpolation is just to add kernel integral operator layers (eq 3) at the beginning and the end of FNO.

**Strength And Weaknesses:**

Strength: the work is well-motivated and clearly written. It has a complete set of numerical experiments which show significant improvements. Also the paper has a careful ablation study.

Weakness: using interpolation to address the mesh points dataset is not very new. Previous works have directly applied interpolation and used FNO as in (https://arxiv.org/abs/2111.05512). As shown in Table 3, the learned interpolation is not very significant from the plain Gaussian interpolation baseline.

Other comments
- In section 4.3, NFS outperforms FNO on the equispaced problem. This is somewhat counter-intuitive. I wonder if the improvement on equispaced problems attributes to differences in architecture such as channel mixing and layernorm, etc.
- A recent work (Geo-FNO) proposes a similar architecture with a different theoretical formulation (https://arxiv.org/abs/2207.05209). Is it possible formulate the kernel interpolation as a deformation?

**Summary Of The Paper:**

The paper extends the Fourier neural operator to non-equispaced setting. It adds interpolation layers at the beginning and the end of FNO. Numerically, the proposed NFS model is more flexible than FNO, while more efficient than graph-based method.

**Summary Of The Review:**

Overall, I think this paper brings a solid contribution. I think the paper is slightly above the threshold.

---

> ### Author Response · Authors · 2022-11-05
> **Response to Reviwer Gvhj**
>
> We express our sincere gratitude for your recognition of our work.
>
> Here are some responses to your concerns:
>
> Firstly, on `Novelty`:
>
> We acknowledge that using interpolation on non-equispaced points to equispaced grids is intuitive and natural. However, we argue that it should not be the fatal weakness of the work.
> The idea of our article is consistent with the procedures of identifying problems and solving them, as
> ---------------------------------------------------
> * Phenomenon 1: Graph-based methods cannot capture the evolution of complex systems, while vision-based ones are able.
> * Phenomenon 2: FNO is the first ML-based method to succeed in modeling NS equations.
> * Explanation: FNO is also a vision-based method, as a member of vision mixers.
> * Solution: Using vision mixers’ architectures which have been explored clearly to improve FNO’s expressivity further.
> ---------------------------------------------
> * Problem 1: Vision Mixers cannot generalize to non-equispaced points.
> * Problem 2: Vision Mixer’s architectures would disable FNO’s mesh-invariance ability.
> * Solution 1: Using Non-equispaced Fourier Transforms to enable FNO’s generalization problem, in which the classical methods use inflexible basis functions (Lagrange or Gaussian).
> * Solution 2: Based on Solution 1, an adaptive interpolation layer is proposed to (i) improve the expressivity; (ii) interpolate to fixed resampled grids, as a solution to both Problem 1 and 2.
> ----------------------------------------------------------
> Therefore, we use the idea of plain interpolation throughout the context to describe NFS, so that the readers can understand the motivation and the nature of the method in a logical way. Moreover, Sec. 3.3 exists with the intention that the readers can figure out why such a simple interpolation layer is effective since it can be used as a derivative of past effective works. We hope that this Computer-Vision style of writing will make the reader think that the Neural Operator is not a very advanced and inaccessible field, and will also attract the interest of researchers in the CV field to advance progress in Neural Operators. We hope the easy-to-understand writing style with extensive experiments to be NFS’s advantage, and more in line with the purpose of ICLR.
>
> Response to other comments:
> 1. Differences between NFS and FNO-Itp (https://arxiv.org/abs/2111.05512):
>
>     + (i) The Fourier interpolation operator in FNO-Itp uses Lagrange interpolation trigonometric polynomials to handle non-equispaced data, while NFS uses adaptively one with universal approximation expressivity.
>
>     + (ii) The improvements in architecture bring large improvements in NFS (Table 3. Flex w/wo LN).
>
> 2. Comparison in Table 3:
> Its fitting ability outperforms Gaus+LN by a relatively small margin. One of the reasons is that the parameters in Gaussian are also learnable, compared with previous methods. However, the ability of **NFS to generalize to unseen meshes surpasses significantly**, which is a very important evaluation aspect in Neural Operators.
>
> 3. NFS outperforms FNO:
> The architecture contributes greatly as you conjectured. In the ablation study, we decoupled the architecture, where LayerNorm is not employed (Sec.4.3 and Table.3). As a result, MAE reaches 5.67 in NS equations, compared with the NFS (MAE=3.27), the performance degenerates significantly. **It is expected when all the tricks in vision mixers’ architecture are removed, the combination of Interpolation+FNO would perform worse than FNO.**
>
> 4. Differences and Similarities between Geo-FNO (https://arxiv.org/abs/2207.05209) and NFS:
>
>     * Similarity: As two contemporaneous pieces of work, the ideas are indeed very similar. Geo-FNO extends FNO to manifolds that are diffeomorphic to Euclidean space. By this mean, it uses a neural network $\phi$ as coordinate transform, to map the target manifold to regular grids in Euclidean space for further FNO architecture. In this way, $\phi$ can be regarded as an interpolation operator on resampled grids in NFS, for further equispaced vision mixer architecture. Besides, the original sampling domain $D$ must be homomorphic to Euclidean space in both papers.
>
>     * Differences:
>        (i) Geo-FNO does not propose contributions to improvements in architecture.
>        (ii) Geo-FNO targets samples on manifolds, so they conduct experiments on Elasticity and Plasticity materials, and Fluid Mechanics in Airfoil. In contrast, NFS focus on non-equispaced sampling, such as irregularly distributed meteorological stations, or cosmic observation data, so we focus on PDEs such as NS equations in rectangle domains.
>
> We still appreciate your recognition of this work and hope that this response will explain why it is written in a plain way with the idea of Interpolation, for the reader to better understand.

---

> > ### Comment · Reviewer_Gvhj · 2022-11-06
> > **comments**
> >
> > Thank the authors for the quick and detailed response. I think we agree on the novelty of the interpolation. It's limited but not fatal. I have some further comments:
> >
> > - The authors mention the contribution of architecture, but I don't think adding LayerNorms is a contribution, but rather hyper-parameter tuning. The original FNO paper presents a simple and generic model. People have used LayerNorms and other tricks in specific tasks (e.g. FNO-transformer https://arxiv.org/abs/2111.13587, FNO++ https://arxiv.org/pdf/2111.13802.pdf).
> > - I think the main contribution of NFS is to have the learned interpolation kernel, which has the same formulation of the kernel integral operator (eq.3), proposed in the Graph kernel operator (GNO). Indeed, if one simply adds GNO layers at the beginning and the end of FNO, it becomes NFS. The contribution of NFS is implementation-wise: since it's non-equispaced  -> equispaced, or equispaced -> non-equispaced, this kernel can be done without graphs. A similar implementation can be found in the Meshfree Flownet https://arxiv.org/abs/2005.01463
> > - NFS vs Geo-FNO. There are three aspects: mesh, geometry, and topology. In my opinion, Geo-FNO has a wider/ more interesting consideration: it can be applied on both non-equispaced meshed, and non-squares geometries, while NFS can only do the first one. But of course, there is no free lunch. If the problem already has a square domain, NFS will be more efficient compared to Geo-FNO. So there is no need to directly compare the two.
> > - Another question: in the paper, the error metric is mainly MAE, while in many other works the metric is relative L2 (RMSE). Do they share a similar behavior?
> > - It's nice to see the DeepONet example. It will be great to put them in the same table in the paper (with NFS, FNO, etc).

---

> > > ### Author Response · Authors · 2022-11-07
> > > **Response**
> > >
> > > We find that your primary concern is that the contribution of this paper may not qualify as an ICLR paper. Here we would like to list the contributions in two aspects:
> > > Methods:
> > > * Extension to non-equispaced scenarios by adaptive interpolation layers (Sec. 3.2), with universal approximation theorem proved (Theorem 3.1).
> > > * Improved architecture including LayerNorm, 2-layer-MLP after Channel Mixing, etc. The architecture enables better performance but will disable mesh-invariance unless interpolation layers are used.
> > > * Further analysis on other variants of interpolation layers in previous works. (Sec.3.3)
> > >
> > > Experiments:
> > > * Promising accuracy compared with Benchmarks in extensive experimental settings, in four PDEs with different parameters. (Sec. 4.2)
> > > * Good generalization ability to unseen meshes. (Table. 3)
> > > * Deep ablation study to figure out the reasons for performance gains. (Sec. 4.3)
> > > * Exploration of non-equispaced Vision Mixers. (Sec. 4.4 and Appendix B.7)
> > > * Representation states analysis to validate the theorem. (Appendix C)
> > > * All Vision Mixers with FNO, NFS are merged into one single code framework, for better comparison and future research convenience of the community. (Appendix B.1)
> > >
> > > We hope the list of contributions can convince you of a qualifying paper as ICLR, thus getting your better recommendation :)
> > >
> > > Response to other comments:
> > > * v.s. Geo-FNO: Two contemporaneous pieces of work use similar formulations, while the motivations of real-world applications are different, leading to two descriptions (from non-equispaced interpolation and manifolds). By this means, we cannot tell the relative merits, both in methods and in experiments due to different equations and so on, but it is another interesting perspective :)
> > > * v.s. Meshfree Flownet: The architectures are similar, which all consist of non-equispaced layers at the beginning or in the end. While Meshfree Flownet uses MLP's outputs with interpolation weights (TIPs: it is confused how the weights are obtained such that mesh-invariance can be assured), we use interpolation layers as a family member of GNO.
> > > * RMSE presentation: In the appendix as the full experimental details, they are all reported, including Table. B.3 ~ B.10.
> > > * The newly updated version includes the experimental results. Thanks for your advice.

---

### Author Response · Authors · 2022-11-06
**General Response to All**

General Response:
Thanks for your valuable comments.
Here are responses to general doubts about our methods.

On comparison with DeepONet:

[Table1:Supplementary results of MAE on two variants of DeepONet]
| PDE     | n_t | r    | n'_t | DeepONet-UNet | DeepONet-Vit |
|---------|-----|------|------|---------------|--------------|
| Burgers | 10  | 512  | 10   | 0.4471        | 0.4782       |
| Burgers | 10  | 512  | 40   | 1.9624        | 2.1707       |
| Burgers | 10  | 1024 | 20   | 0.6541        | 1.6130       |
| Darcy   | 1   | 64   | 1    | 0.3753        | 0.5119       |
| Darcy   | 1   | 128  | 1    | 0.9488        | 0.9614       |
| Darcy   | 1   | 256  | 1    | 0.9692        | 1.3216       |
| NS      | 10  | 64   | 10   | 3.4436        | 3.9745       |
| NS      | 10  | 64   | 40   | 10.2950       | 12.3314      |
| NS      | 10  | 128  | 20   | 7.1394        | 9.3471       |

Since vanilla DeepONet uses MLP as Branch Net, it cannot be implemented in such a high-resolution dataset, because for a resolution like the trial (NS $n_s=4096, n_t=10, n’_t=10$), DeepONet assigns each data point a weight parameter in a single MLP, leading the MLP’s parameter number reaches $O(n^2_sn^2_t) \approx 40960^2$ in a single Branch Net, which is infeasible in practice. In the original paper, the spatial point’s number in the experiments is set as 40, far less than in the recent Neural Operator’s evaluation protocol.

One feasible alternative is to use other architecture to replace the original MLP, thus allowing DeepONet to handle high-resolution data. For example, CNN and Vit. Therefore, we here conduct further experiments on the three equations in the context, to evaluate DeepONet-UNet and DeepONet-Vit as two variants of vanilla DeepONet for comparison. Note that the architecture of variants of DeepONet are all limited to equispaced data. The detailed architecture will be added in the Appendix in the newly updated version.

On training and test size:

  Here we list the training set sizes and test set sizes as the experimental protocols used in benchmark Neural Operators and ours in Table 2. Besides, we generated more samples and conducted experiments to give the metrics in Table.3 on different test sizes in the three equations.


[Table2: Training and testing sample numbers in Benchmarks]
| Methods  | #Training | #Test  | Equation                               |
|----------|-----------|--------|----------------------------------------|
| DeepONet | 10000     | 100000 | Gravity pendulum                       |
| FNO      | 1000      | 200    | Navier Stokes/Burger's/Darcy           |
| FNO      | 10000     | 200    | Navier Stokes                          |
| MGKN     | 1000      | 200    | Burger's/Darcy                         |
| FairComp | 1000      | 200    | PWC/Cont/Advction II&III/Navier Stokes |
| FairComp | 1900      | 100    | Darcy                                  |
| FairComp | 15        | 2      | Electroconvection                      |
| FairComp | 100       | 50     | Euler                                  |
| FairComp | 28        | 20     | Airfoil                                |
| NFS |  960 | 240 | Navier Stokes/Burger's/Kdv/Darcy    |

MGKN is Multiple Graph Kernel Operator (https://proceedings.neurips.cc/paper/2020/file/4b21cf96d4cf612f239a6c322b10c8fe-Paper.pdf) ;
FairComp is the fair empirical comparison. (https://arxiv.org/abs/2111.05512), and see more details in Table S.1. in the paper.

Besides, some supplementary experiments are conducted to satisfy one reviewer's curiosity.

[Table3: MAE on different test sizes on NS equations]
| PDE | n_t | r   | n'_t | test_size | MAE    |
|-----|-----|-----|------|-----------|--------|
| NS  | 10  | 64  | 10   | 500       | 0.8621 |
| NS  | 10  | 64  | 40   | 500       | 3.1176 |
| NS  | 10  | 128 | 20   | 500       | 1.8365 |
| NS  | 10  | 64  | 10   | 1000      | 0.8624 |
| NS  | 10  | 64  | 40   | 1000      | 3.1159 |
| NS  | 10  | 128 | 20   | 1000      | 1.8378 |

Revision of the latest version of the paper:
* Add Appendix C on representation state analysis corresponding to Theorem 3.1. Add a short note before Theorem 3.1 for reference to Appendix C.
* Add DeepONet-U(UNet) and DeepONet-V(Vit) as two variants of feasible DeepONet to benchmark comparisons. Add the implementation of the two methods in Appendix B.1.

---

### Author Response · Authors · 2022-12-01
**Request to responses**

Dear Reviewers,

Hope you are doing well.

We have posted our response to the raised questions and your concerns. Also, additional experiments are conducted to evaluate our method more thoroughly and revised our paper accordingly.

Could you take a look at the response and the revision, and let us know if there are any further feedback? Thank you!

Best,

The authors

---

### Decision · Program_Chairs · 2023-01-20

**Decision:**

Reject

**Justification For Why Not Higher Score:**

The revised paper continues to have issues with novelty and clarity of writing, and therefore may benefit from extensive revisions.

**Justification For Why Not Lower Score:**

NA

**Metareview: Summary, Strengths And Weaknesses:**

The paper proposes a new neural PDE solver that they call "Non-equispaced Fourier PDE Solver", or NFS. Addressing irregularly sampled data is important in practice, but current neural PDE approaches typically sweep this issue under the rug. The authors tackle this by leveraging non-equispaced FFT layer (with a learned interpolation kernel) before and after several standard Fourier Neural Operator (FNO) layers. Along the way, the authors also point out connections to ViT-type architectures and prove universal approximation results. Simulation results on benchmark PDEs (Burgers, Darcy, Navier-Stokes) are also provided.

The reviewers highlighted a few positive points about the paper. Overall, the idea of interpolating non-equispaced points is intuitive, and the improvements over baselines using NFS was quite clear from the experiments.

However, several negative points were also raised. Questions about novelty persisted. As one reviewer pointed out, the use of interpolation kernels in the beginning and end can be viewed as a special case of adding a kernel integral operator to FNO; the only novelty here was the aspect of using learned interpolators. Comparisons to more baselines (particularly, DeepONets) were missing. In my personal reading of the paper, I also found that the authors may have tried to cram too many ideas (possibly hampered by tight ICLR page limits) -- for example, sections 2.4 and 2.5 could be safely removed without changing overall impact, and the freed up space could be used to improve clarity and address novelty concerns.

The authors (in some cases, forcefully) responded to the initial set of reviews. In particular, additional comparisons to DeepONets were provided. However, the original concerns (about novelty and clarity) still remain, and the paper can benefit from one more round of extensive revisions.



**Summary Of Ac-Reviewer Meeting:**

NA